# Statistical Estimation of Resistance to Cyclic Deformation of Structural Steels and Aluminum Alloy

**Žilvinas Bazaras and Vaidas Lukoševičius \***

Faculty of Mechanical Engineering and Design, Kaunas University of Technology, Studentų Str. 56, 51424 Kaunas, Lithuania; zilvinas.bazaras@ktu.lt
\* Correspondence: vaidas.lukosevicius@ktu.lt

**Abstract:** Resistance to cyclic loading is a key property of the material that determines the operational reliability of the structures. When selecting a material for structures operating under low-cycle loading conditions, it is essential to know the cyclic deformation characteristics of the material. Low-cycle strain diagrams are very sensitive to variations in chemical composition, thermal processing technologies, surface hardening, loading conditions, and other factors of the material. The application of probability methods enables the increase in the life characteristics of the structures and the confirmation of the cycle load values at the design phase. Most research papers dealing with statistical descriptions of low-cycle strain properties do not look into the distribution of low-cycle diagram characteristics. The purpose of our paper is to provide a probability assessment of the low-cycle properties of materials extensively used in the automotive and aviation industries, taking into account the statistical assessment of the cyclic elastoplastic strain diagrams or of the parameters of the diagrams. Materials with contrasting cyclic properties were investigated in the paper. The findings of the research allow for a review of durability and life of the structural elements of service facilities subjected to elastoplastic loading by assessing the distribution of low-cycle strain parameters, as well as the allowed distribution limits.

**Keywords:** cyclic loading curves parameters; durability; log-normal distribution; low cycle fatigue; probability; normal distributions; Weibull distribution

## 1. Introduction

Transport, engineering, shipbuilding, energy, metallurgy, chemicals, and other industries have a major impact on economic development. To achieve more rapid economic growth, it is necessary to make the most of the available equipment production capacity. Problems are encountered, such as increasing the durability of structural elements in terms of strength criteria and reducing the cost of metal. The continuous growth of machine production capacities, speeds, lifting weights, and other parameters is associated with an accumulation of stresses in the structural elements. In many areas of production, machines are subjected to high loads. This generates cyclic elastic-plastic deformation in the individual structural elements. It is important to design strong, cost-effective, and reliable structures. This can be done in the development process using the latest developments in design and computation.

Statistical probabilistic methods are used to improve the reliability and safety of a structure or product during its lifetime. The basic patterns of fatigue resistance are analyzed in terms of the probabilistic distribution of independent variables, allowing the basic characteristics of durability to be assessed using statistical analysis.

The probabilistic prediction of service life and fracture reliability involves the determination of allowable stresses depending on design factors, conditions, and loads. The reliability and endurance of the structure are affected by the dispersion of the number of durability cycles, random load deviations, and long-term loading of the stresses, which causes fatigue failures. Variations in the characteristics of the fatigue curve reflect the

heterogeneity of the material, which depends on metallurgical and technological factors in the manufacturing process, as well as the effect of external factors during service [1,2].

In recent years, a considerable number of methods of probabilistic analysis of low cycle properties have been investigated [3–8]. Cyclic stress–strain curves, the same as uniaxial stress–strain curves, are known to be very sensitive to variations in material chemical composition, heat treatment, surface hardening, loading conditions, and other factors. As a result, in low-cycle experiments, there is a significant variation in the cyclic deformation characteristics, in many cases significantly exceeding the variation of mechanical characteristics. Computations of low-cycle fatigue life are performed using the parameters of cyclic deformation curves, but so far only a few statistical estimates of these parameters have been made.

Strzelecki [9] presents the proposed characteristics of the *S-N* curve using the two- and three-parameter Weibull distribution for fatigue limit and limited life, respectively. The parameters of the proposed model are estimated using the maximum likelihood method. Additionally, a method to estimate initial values for the likelihood function has been presented. Correia et al. [10] have proposed a generalization of the probabilistic fatigue model of Castillo et al. [11] and investigated how fatigue models could be obtained in certain cases. Marohnic et al. [12] have proposed an investigation of the estimation of cyclic stress–strain curves, namely, direct estimation from monotonic properties and indirect estimation where relations between monotonic properties and points on cyclic stress–strain curves are identified, and new values obtained. Xiang et al. [13], Xu et al. [14] have proposed a general probabilistic life prediction methodology for accurate and efficient fatigue life prediction. Williams et al. [15] have presented a method for the development of accurate statistical strain-life curves from experimental data from strain-controlled uniaxial fatigue tests. Gu et al. [16] have developed an estimation of interval limits based on strain-life fatigue data, new strain-life lines have been fitted, and the expansion root mean square error has been introduced as the index of grouped data points. Sakin et al. [17] have investigated bending fatigue behaviors in glass fiber reinforced polyester composite plates made from woven rope with four different weights, random distributed glass material with different weights, and polyester resin; also, the S-N curves have been introduced to identify the first failure time as reliability and safety limits for designers' benefits. Haidyrah et al. [18] have studied the fatigue properties of grade 91 ferritic–martensitic steel using a mini-specimen (Krouse type) suitable for reactor irradiation studies of nuclear materials, and have developed the S-N curve and analyzed the mean fatigue life using the mean fatigue life. Liu et al. [19] have investigated the reliability of low-cycle fatigue life using finite element analysis as a numerical experiment tool and have developed a simulation method to obtain the probability density distributions of the stress level and strain level at dangerous points in the disc structure of an aeronautic engine turbine. Khelif et al. [20] compared different statistical methods and distributions to provide a convenient modeling of the results of the tensile and fatigue tests of commercially available polyethylene compression-molded sheets. They found that while the three-parameter Weibull and log-normal types are suitable for lifetime prediction, the two-parameter Weibull distribution is more conservative for probabilistic fatigue design.

Zhu et al. [21] have aimed to develop a Bayesian framework for the probabilistic prediction of the low cycle fatigue life and quantifying the uncertainty of material properties, the total uncertainty of the input sand model resulting from the choice of different deterministic models in an LCF regime. Zhao et al. [22] have analyzed the statistical evolution of small-fatigue cracking behavior, where three considerations are given: total fit, consistency of statistical parameters with test data, and the practice of commonly used distributions, namely Weibull (two- and three-parameter), normal, log-normal, extreme minimum, extreme maximum, and exponential. Sun et al. [23] have proposed a statistically consistent fatigue damage model under constant and variable amplitude loads based on the linear Miner rule, which allows a quantitative depiction of the probabilistic properties of fatigue damage and fatigue life.

Daunys et al. [24,25] have investigated the dependence of low cycle fatigue life on the mechanical properties of joint steels welded in nuclear power plants. Raslavičius et al. [26] investigated the low cycle durability of the primary nozzles of WWER reactors (Water–Water Energetic Reactor) of nuclear power plants made of steels 22 k and 15Cr2MoVA. The study describes the specific case of the 15Cr2MoVA steel defective housing of the nozzle made of steel 15Cr2MoVA and after crack repair along the length of the diameter of the nozzle. The analysis of the welded specimens showed good agreement between the empirical and theoretical curves and can be used in the lifetime and fatigue life predictions for WWER pressure vessels still applicable in the power generation industry. Bazaras et al. [27,28] have investigated the probabilistic low-cycle fatigue life and the dependence of low-cycle durability on the mechanical properties of the WWER nuclear power plant reactor of 15Cr2MoVA steel and structural steel C45.

The literature review suggests that there are not yet any consistent studies on the deformation parameters of low-cycle fatigue. Most papers concerned with the statistical description of the deformation characteristics of low-cycle fatigue have done so only by calculating the durability to crack initiation or complete failure. The statistical description of the parameters of the deformation diagrams and of the fatigue curves of the low-cycle fatigue was not considered at all.

Based on the topics discussed above, the main outcomes of this paper are as follows: (1) we determine the dependence of the parameters of the low-cycle fatigue curves on the type of load and material properties; (2) we perform a statistical evaluation of the parameters of the low-cycle loading curves for structural materials; (3) we confirm the agreement between the hypotheses of the empirical distribution and the theoretical law of normal distribution according to compatibility criterion; and (4) we present a comparison of the analytical low-cycle fatigue probability curves and the experimental data.

## 2. Materials and Methods

### 2.1. Experiment and Materials

Low-cycle tension-compression fatigue experiments were carried out under controlled stress ($\sigma_k = constant$) conditions. The experimental equipment used for fatigue tests consisted of a 50 kN testing machine Instron 8801 (Norwood, MA, USA) series Servo Hydraulic Fatigue Testing System with FastTrack 8800 controller and the electronic device that was designed to record the stress–strain curves. A deformation rate of four cycles per minute was used for fatigue tests. Mechanical characteristics were measured with an error not exceeding ±1% of the deformation scale. For dynamic inertia compensation during fatigue testing, Dynacell Dynamic Load Cell ± 250 N was used (according to ISO 75001/1 Class 0.5, ISO 10,002 Part 2, ASTM E4, EN10002 Part 2 and JIS (B7721, B7733).

Fatigue tests have been performed according to the GOST 25,502-79 standard (Strength analysis and testing in machine building. Methods of metals mechanical testing. Methods of fatigue testing.) [29]. Standard GOST 22,015-76 (Quality of product. Regulation and statistical quality evaluation of metal materials and products on speed-torque characteristics.) [30] was used to calculate the statistical characteristics.

The specimens used for the cyclic deformation tests under the linear stress state shall ensure a homogeneous stress state in the test piece until a fatigue crack occurs. The short test section in cylindrical specimens is used to prevent the specimen from buckling under large compressive strains. The quality of the surface of the parts to be tested in the test specimens is very important for durability. Figure 1 shows a drawing of the sample that best meets these conditions.

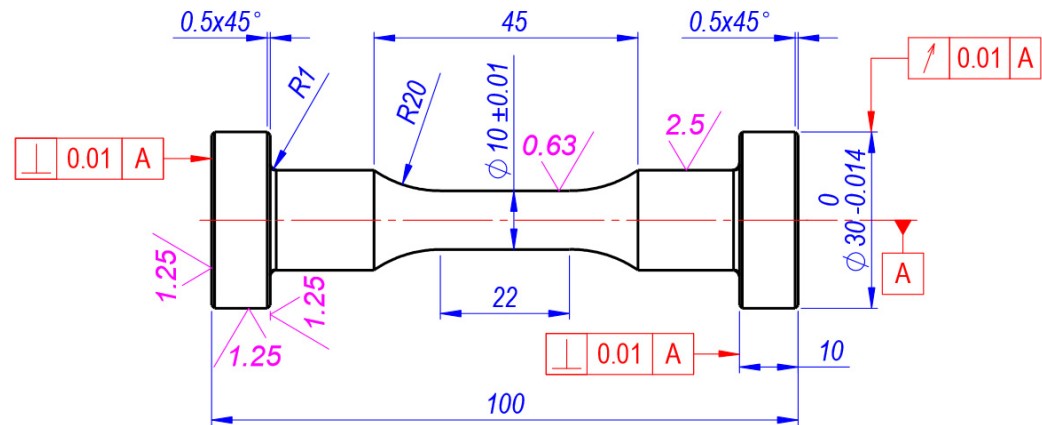

**Figure 1.** Samples of circular cross section for low-cycle tension-compression fatigue experiments (Units in mm).

Tables 1 and 2 show the mechanical properties and chemical composition and of cyclically softening (alloyed steel 15Cr2MoVA), stable (structural steel C45), and hardening (aluminum alloy D16T1) materials.

**Table 1.** The chemical composition of the materials.

| Material | C | Si | Mn | Cr | Ni | Mo | V | S | P | Mg | Cu | Al |
|---|---|---|---|---|---|---|---|---|---|---|---|---|
| | % | | | | | | | | | | | |
| 15Cr2MoVA (GOST 5632-2014) | 0.18 | 0.27 | 0.43 | 2.7 | 0.17 | 0.67 | 0.30 | 0.019 | 0.013 | - | - | - |
| C45 (GOST 1050-2013) | 0.46 | 0.28 | 0.63 | 0.18 | 0.22 | - | - | 0.038 | 0.035 | - | - | - |
| D16T1 (GOST 4784-97) | - | - | 0.70 | - | - | - | - | - | - | 1.6 | 4.5 | 9.32 |

**Table 2.** Mechanical properties of the materials.

| Material | $e_{pr}$ | $\sigma_{pr}$ | $\sigma_{0.2}$ | $\sigma_u$ | $S_k$ | $\psi$ |
|---|---|---|---|---|---|---|
| | % | | MPa | | | % |
| 15Cr2MoVA (GOST 5632-2014) | 0.200 | 280 | 400 | 580 | 1560 | 80 |
| C45 (GOST 1050-2013) | 0.260 | 340 | 340 | 800 | 1150 | 39 |
| D16T1 (GOST 4784-97) | 0.600 | 290 | 350 | 680 | 780 | 14 |

The mechanical characteristics required for the investigation have been generated by experiments on the specimens made of three materials: alloyed steel 15Cr2MoVA (160 specimens), structural steel 45 (220 specimens), aluminum alloy D16T1 (120 specimens). Different number of samples was used to determine the effect of the sample size on the statistical results. All test equipment and methods are explained in detail in the literature [31].

### 2.2. Low Cycle Deformation Diagram and Its Characteristics

The diagram illustrated below (Figure 2) is based on the experimental results. The diagram for each semi-cycle loading forms a general curve, independently of the loading level [31]. Figure 2 shows four semi-cycle diagrams for symmetric loading for stress-limited loading ($\sigma = constant$). The diagram of the initial semi-cycle (OA) is presented in coordinates ($\sigma - e$) and the coordinates of the first semi-cycle (curve AB) and all other semi-cycles are ($S - e$), starting at the elastic part of each semi-cycle (at the start of the unloading). The zero semi-cycle is characterized by the initial stress $\sigma_0$ and the corresponding initial strain $e_0$, the stress $\sigma_{pr}$ and strain $e_{pr}$ of material proportionality limit. The diagram of each subsequent semi-cycle is characterized by the stress $S$, the strain $\varepsilon$, the cyclic proportionality limit $S_T$, and the width $\delta_k$ of the semi-cycle hysteresis loop.

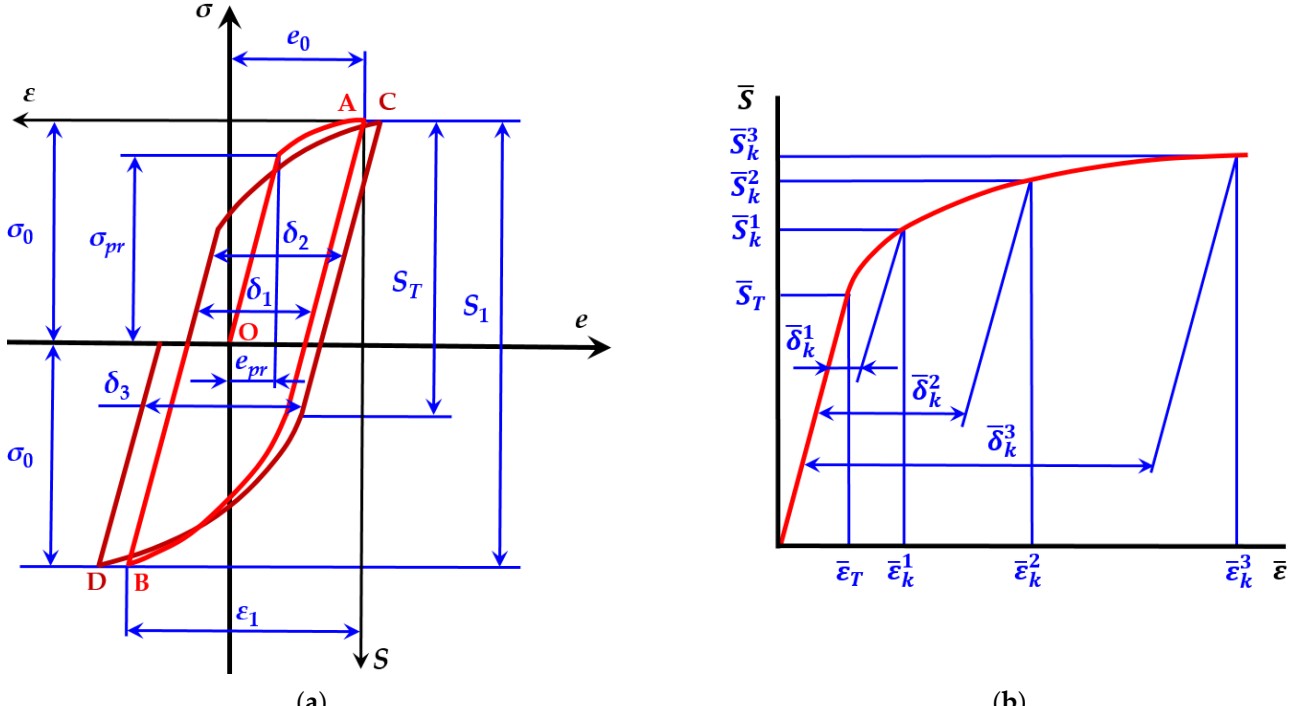

**Figure 2. (a)** Low cycle deformation diagram; **(b)** relative $k$ semi-cycle deformation diagram.

The parameters of the cyclic strain diagram are expressed in relative units to facilitate the analysis, that is, in relation to the limits of proportionality:

$$\bar{e}_0 \;=\; \frac{e_0}{e_{pr}}, \; \bar{\sigma}_0 \;=\; \frac{\sigma_0}{\sigma_{pr}}, \; \bar{\varepsilon}_k \;=\; \frac{\varepsilon_k}{e_{pr}}, \; \bar{S}_k \;=\; \frac{S_k}{\sigma_{pr}} \text{ etc.} \tag{1}$$

To describe the analytical relationship between cyclic stresses and strains in an elasto-plastic region (reference coordinates $S - \varepsilon$), the dependence proposed by Serensen and Shneiderovich [32] is shown:

$$\begin{cases} \bar{\varepsilon}_k \;=\; \bar{S}_k, \bar{S}_k \leq \bar{S}_{Tk} \\ \bar{\varepsilon}_k \;=\; \bar{S}_k + \bar{\varepsilon}_{pk}, \bar{S}_k > \bar{S}_{Tk} \end{cases}. \tag{2}$$

It was assumed that $\bar{\varepsilon}_{pk} \;=\; \bar{\delta}_k$, then from the Equation (2) and Figure 2b the following can be written:

$$\bar{\varepsilon}_k \;=\; \bar{S}_k + \bar{\delta}_k. \tag{3}$$

It has been found [32] that the width of the hysteresis loop $\bar{\delta}_k$ depending on the number of loading semi-cycles $k$, forms a straight line in the coordinates $lg\bar{\delta}_k - lgk$, so that:

$$lg\bar{\delta}_k \;=\; \alpha \cdot lgk + lg\delta_1. \tag{4}$$

After finding the logarithms, can be written:

$$\bar{\delta}_k \;=\; \bar{\delta}_1 k^\alpha \text{ or } \bar{\delta}_k \;=\; \bar{\delta}_1 F(k). \tag{5}$$

The dependence on the initial semi-cycle strain $\bar{e}_0$:

$$\bar{\delta}_1 \;=\; A_1\left(\bar{e}_0 - \frac{\bar{S}_{Tk}}{2}\right). \tag{6}$$

The function of the number of semi-cycles:

$$\text{for cyclically hardening material}: \ F(k) \ = \ \frac{1}{k^\alpha}, \tag{7}$$

$$\text{for cyclically softening material}: \ F(k) \ = \ \exp\beta(k-1), \tag{8}$$

$$\text{for cyclically stable material}: \ F(k) \ = \ 1, \alpha \ = \ \beta \ = \ 0.$$
$$\text{Where } \beta \ = \ c\left(\overline{e}_0 - \frac{\overline{S}_{T1}}{2}\right). \tag{9}$$

Using Equations (2), (5) and (6), for under loading with controlled stress $(\sigma_k \ = \ constant)$ it can be written:

$$\text{for cyclically hardening material}: \ \overline{\varepsilon}_k \ = \ \overline{S}_k + A_1\left(\overline{e}_0 - \frac{\overline{S}_{Tk}}{2}\right)\frac{1}{k^\alpha}, \tag{10}$$

$$\text{for cyclically softening material}: \ \overline{\varepsilon}_k \ = \ \overline{S}_k + A_1\left(\overline{e}_0 - \frac{\overline{S}_{Tk}}{2}\right)k^\alpha, \tag{11}$$

$$\text{for cyclically stable material}: \ \overline{\varepsilon}_k \ = \ \overline{S}_k + A_1\left(\overline{e}_0 - \frac{\overline{S}_{Tk}}{2}\right). \tag{12}$$

For under loading with controlled strain $(e_k \ = \ constant)$, the following can be written:

$$\text{for cyclically hardening material}: \ \overline{S}_k \ = \ \overline{\varepsilon}_k + A_1\left(\overline{e}_0 - \frac{\overline{S}_{Tk}}{2}\right)\frac{1}{k^\alpha}, \tag{13}$$

$$\text{for cyclically softening material}: \ \overline{S}_k \ = \ \overline{\varepsilon}_k + A_1\left(\overline{e}_0 - \frac{\overline{S}_{Tk}}{2}\right)k^\alpha, \tag{14}$$

$$\text{for cyclically stable material}: \ \overline{S}_k \ = \ \overline{\varepsilon}_k + A_1\left(\overline{e}_0 - \frac{\overline{S}_{Tk}}{2}\right). \tag{15}$$

From the equations presented above it can be seen that the following parameters are used to analytically describe the cyclic deformation diagrams: $\overline{S}_{Tk}, \ A_1, A_2, \ \alpha$. The authors' investigations showed that the cyclical proportionality limit increases slightly for hardening materials, decreases for softening materials, and remains constant for stable materials. As the variation of the values is small, it is generally assumed that the limit of proportionality is equal to the limit of the first semi-cycle, i.e., $\overline{S}_T \ = \ \overline{S}_{T1} \ = \ \overline{S}_{T2}\ldots\overline{S}_{Tk} \ = \ constant$. The parameters $A_1$ and $A_2$ define the loop widths $\overline{\delta}_1$ and $\overline{\delta}_2$ of the first and second semi-cycles, respectively. The parameter $\alpha$ describes the cyclic properties of materials, i.e., hardening, softening, and stability.

The scientific studies carried out by the authors [33] of strain-limited loading under torsion conditions have shown satisfactory agreement between strain-limited and stress-limited loading curves. Based on this experimental fact, using Equation (3) for strain-limited loading, the following can be written:

$$\overline{S}_k \ = \ \overline{\varepsilon}_k - \overline{\delta}_k. \tag{16}$$

For a cyclically anisotropic material, which unequally responds to tensile and compressive stresses and, therefore, accumulates unilateral plastic strain, a graphical representation of the dependence in coordinates $lg\overline{\delta}_k - lgk$, gives two parallel lines (Figure 3). This means that there are two values of $\overline{\delta}_1$: $\overline{\delta}_1$—actual value and $\overline{\delta}_{1f}$—fictitious value. The parameters $A_{1f}$ and $A_{2f}$ define the loop widths $\overline{\delta}_{1f}$ and $\overline{\delta}_{2f}$ of the first and second semi-cycles respectively.

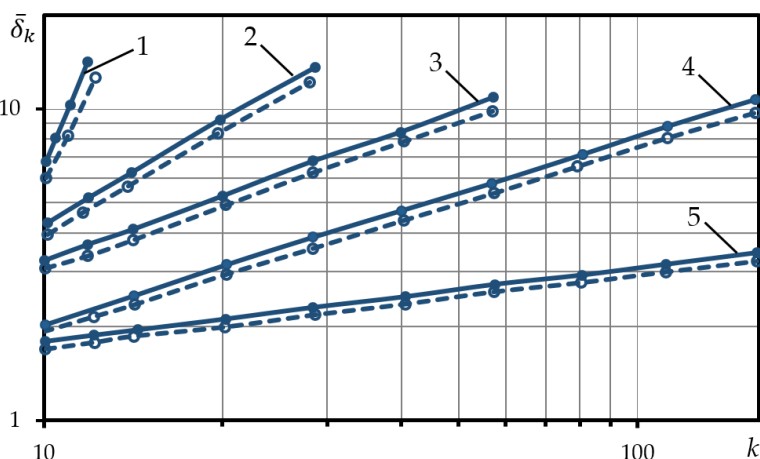

**Figure 3.** Hysteresis loop widths dependence of softening materials on the number of loading semi-cycles (1—$\bar{e}_0 = 9.6$; 2—$\bar{e}_0 = 5.2$; 3—$\bar{e}_0 = 2.4$; 4—$\bar{e}_0 = 1.4$; 5—$\bar{e}_0 = 1.2$ ).

For cyclically anisotropic materials (accumulated plastic strain after loading semi-cycles in the direction of tension) $\bar{\delta}_{k4} > \bar{\delta}_{k4f}$:

$$\bar{\delta}_{k4} = \bar{\delta}_{1f}F(k), \; \bar{\delta}_{k4f} = \bar{\delta}_1 F(k). \tag{17}$$

where $\delta_{1f}$ is the fictitious width of the elastoplastic hysteresis loop determined by the dependence:

$$\bar{\delta}_{1f} = A_2\left(\bar{e}_0 - \frac{\overline{S}_{Tk}}{2}\right). \tag{18}$$

According to Equation (18), constant $A_2$ determines the shape of the deformation diagram of the second semi-cycle.

From Equations (17) and (18), it is obvious that the accumulated plastic strain after loading semi-cycles in the direction of tension is:

$$\bar{e}_{pk} = \bar{e}_0 - \bar{\sigma}_0 + \sum_{1}^{k}(-1)^k\bar{\delta}_k. \tag{19}$$

Substituting the dependence for the hysteresis loop width into Equation (19), is obtained the calculation of the equations for determining the accumulated plastic strain, that is, taking into account Equations (5) and (6) for softening materials, is retrievable the following:

$$\bar{e}_{pk} = \bar{e}_0 - \bar{\sigma}_0 - \left(\bar{e}_0 - \frac{\overline{S}_T}{2}\right)\sum_{1}^{k}\left[\frac{A_1}{2} - (-1)^k\frac{A_1}{2}\right]\exp\beta(k-1) + \left(\bar{e}_0 - \frac{\overline{S}_T}{2}\right)\sum_{1}^{k}\left[\frac{A_2}{2} + (-1)^k\frac{A_2}{2}\right]\exp\beta(k-1). \tag{20}$$

where $k = 2N$ is for even semi-cycles, $k = 2N + 1$ is for odd semi-cycles, or by making the respective substitution in Equation (12) for hardening or stable materials:

$$\bar{e}_{pk} = \bar{e}_0 - \bar{\sigma}_0 - \left(\bar{e}_0 - \frac{\overline{S}_T}{2}\right)\sum_{1}^{k}\left[\frac{A_1}{2} - (-1)^k\frac{A_1}{2}\right]\frac{1}{k^\alpha} + \left(\bar{e}_0 - \frac{\overline{S}_T}{2}\right)\sum_{1}^{k}\left[\frac{A_2}{2} + (-1)^k\frac{A_2}{2}\right]\frac{1}{k^\alpha}. \tag{21}$$

It is notable that Equations (21) and (22) are valid only in the range of semi-cycles, in which the dependences of the hysteresis loop width are valid, i.e., only for uniform plastic deformation along the entire length of the sample. The appearance of the necks on the specimen leads to significant deviations from these dependences. This phenomenon can

be observed particularly for the characteristics of 15Cr2MoVA steel, which fractures mostly quasi-statically in almost the entire range of the investigated strains.

## 3. Results and Discussion

### 3.1. Statistical Assessment of the Low-Cycle Fatigue Curves Parameters

Heat-resistant steel 15Cr2MoVA has been anisotropic during cyclic loading, that is, it accumulates plastic strain in the direction of tension. Structural steel C45 has been a cyclically stable anisotropic material. The D16T1 aluminum alloy has been cyclically hardened during cyclic loading with stress-limited loading and has not unilaterally accumulated plastic deformation.

As follows of the equations from the previous paragraph, the values of parameters $A_1$, $A_{1f}$, $A_2$, $A_{2f}$, $\beta$, $c$, $\overline{S}_T$ and $\alpha$ are necessary in the analytical description of the cyclic deformation curves. According to the methodology provided in [34], the histograms for the cyclic deformation parameters mentioned above were defined (Figures 4 and A1–A3). The histograms of parameter $A_1$ for 15Cr2MoVA steel indicate a significant scatter of the parameter, and with an increase in the volume of the statistical series (the level of loading $\overline{\sigma}_0 = 1.00$—20 samples and $\overline{\sigma}_0 = 1.14$—40 samples), their form approaches the normal distribution. For all loading levels, the histograms of parameter $A_1$ of steel 15Cr2MoVA are shifted to the left, that is, they have a positive asymmetry.

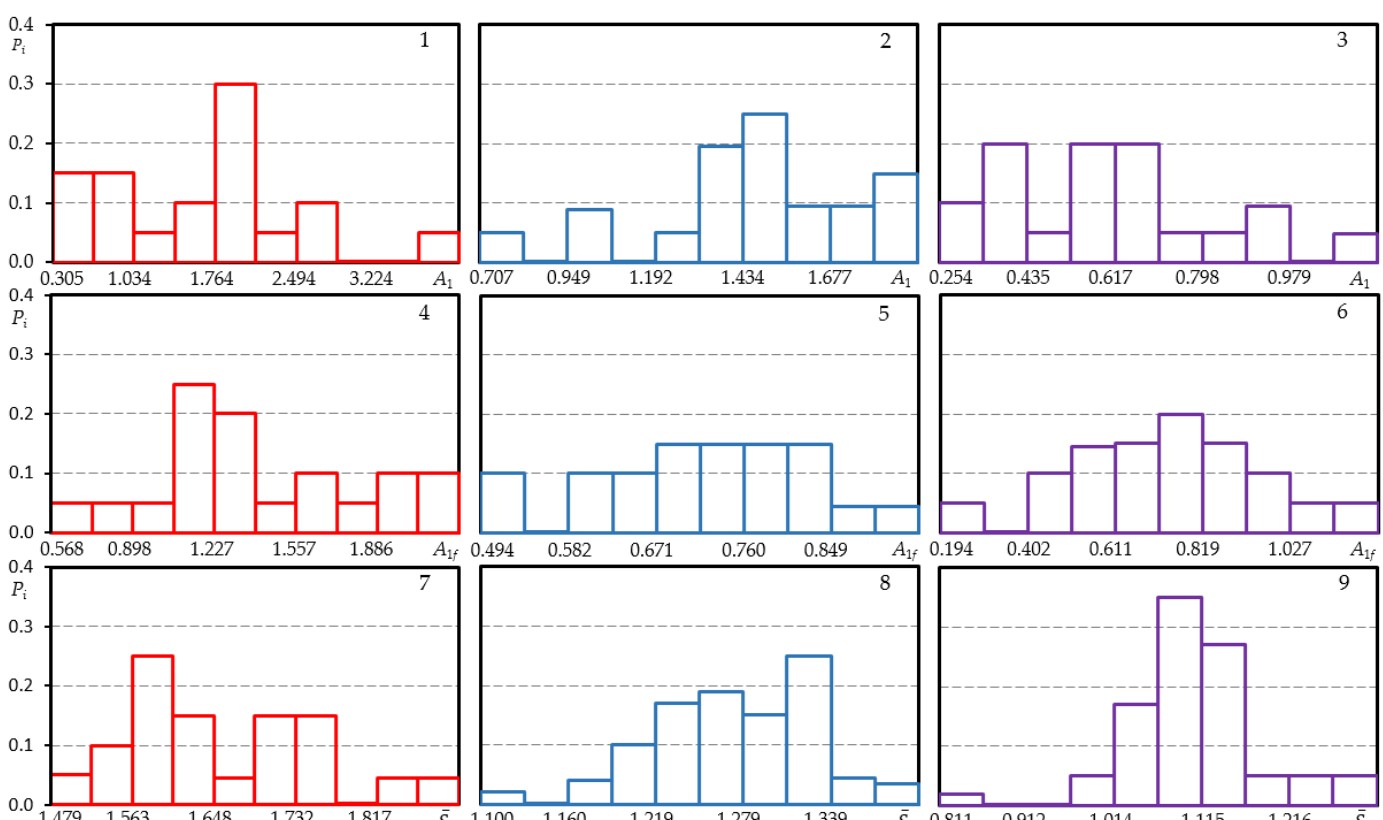

**Figure 4.** Histograms of low cyclic load diagram characteristics of steel 15Cr2MoVa (1, 4, 7), steel 45 (2, 5, 8)—loading level $\overline{\sigma}_0 = 1.00$ and aluminum alloy (3, 6, 9)—loading level $\overline{\sigma}_0 = 1.12$ (**1–3**—$A_1$; **4–6**—$A_{1f}$; **7–9**—$\overline{S}_T$).

The almost same nature of the histograms of parameter $A_1$ is observed for steel C45. For loading levels $\overline{\sigma}_0 = 1.00; 1.25$, there is positive asymmetry, and for loading level $\overline{\sigma}_0 = 1.50$, it is negative.

From the analysis of the parameter distribution histograms for steel C45, it follows that they differ little from the parameter $A_1$ histograms. For this parameter, the sample size also

influences the shape of the histogram, i.e., with increasing samples, the fit to the normal distribution improves. For loading levels $\bar{\sigma}_0 = 1.00; 1.12$, slight negative asymmetry is observed, and for $\bar{\sigma}_0 = 1.00; 1.12$, the asymmetry is positive.

The parameter $A_{1f}$ histograms for steel C45, compared to the histograms for steel 15Cr2MoVA, do not correspond exactly to the normal distribution. Furthermore, a negative asymmetry is observed at loading levels $\bar{\sigma}_0 = 1.00; 1.25$, and at loading level $\bar{\sigma}_0 = 1.50$, the asymmetry is positive.

The parameter $A_{2f}$ histograms are similar to those of parameter $A_{1f}$. For steel 15Cr2MoVA, with $\bar{\sigma}_0 = 1.00$, the asymmetry is slightly negative, for $\bar{\sigma}_0 = 1.12$—negative, while with $\bar{\sigma}_0 = 1.25$—positive. For steel C45, the parameter $A_{2f}$ histograms are identical to those of parameter $A_{1f}$. At loading levels $\bar{\sigma}_0 = 1.00; 1.25$, negative asymmetry is observed, and at level $\bar{\sigma}_0 = 1.50$, the asymmetry is positive.

It should be noted that the histograms for the parameter distribution depend on the loading level and sample size, that is, with an increase in the samples and loading level, the shape of the histograms approaches the normal distribution. With an increase in the loading level for steel 15Cr2MoVA, the quasi-static fracture zone is approached, as more steady change in cyclic deformation curves is established compared to the transient fracture zone, leading to a smaller scatter of parameters reflecting this process. For 15Cr2MoVA steel, the parameter $\beta$ histograms have positive asymmetry at all loading levels.

The parameter $\beta$ is a function of the parameter $c$, consequently the parameter $c$ histograms do not vary much from the parameter $\beta$ histograms (Figure A3).

From the consideration of the histograms for the cyclic proportionality limit $\bar{S}_T$, it follows that, with the increasing sample size, the shape of the histograms approaches the normal distribution. For the 15Cr2MoVA and C45 steels with a sample size of 80 points (Figure 4), the histograms in their shape agree with the normal distribution. The asymmetry of the distribution of the parameter for steel 15Cr2MoVA is negative, while for steel C45, it is positive. For the aluminum alloy D16T1, the sample size of $\bar{S}_T$ is 20 samples (Figure 4), so the histogram does not have a clearly defined distribution law.

From the analysis of the distribution of cyclic deformation parameters $A_1$, $A_{1f}$, $A_2$, $A_{2f}$, $\beta$, $c$, $\bar{S}_T$ and $\propto$, it can be concluded that the shape of histograms mainly depends on the sample size, i.e., the number of tests, and, with an increase $i$ the sample size of the histogram for most of the parameters, the shapes of the histograms approach the normal distribution.

For a more accurate definition of the distribution law, a computer-aided study has been conducted to confirm the agreement between the hypotheses of the empirical distribution and the theoretical law of normal distribution according to the Smirnov compatibility criterion $\omega^2$ [35]:

$$\omega^2 = \frac{1}{12n} + \sum_{i+1}^{n} [W(x_i - \Phi(\hat{z}_i)]^2. \tag{22}$$

The criterion for the results of a sample to satisfy the law of the normal or log-normal distribution is expressed by the inequality:

$$\omega^2 (1 + \frac{1}{2n}) \leq W_{a_1}^2. \tag{23}$$

The results of agreement with the normal distribution according to the Smirnov compatibility criterion $\omega^2$ are provided in Table 3.

**Table 3.** Goodness-of-Fit estimation of cyclic properties that the result samples have a log-normal distribution using the Smirnov compatibility criterion $\omega^2$.

| Cyclic Properties | 15Cr2MoVa | | | | | | | | | | | | C45 | | D16T1 | |
|---|---|---|---|---|---|---|---|---|---|---|---|---|---|---|---|---|
| | $\overline{\sigma}_0 = 1.00$ | | $\overline{\sigma}_0 = 1.12$ | | $\overline{\sigma}_0 = 1.25$ | | $\overline{\sigma}_0 = 1.00$ | | $\overline{\sigma}_0 = 1.12$ | | $\overline{\sigma}_0 = 1.50$ | | $\overline{\sigma}_0 = 1.15$ | | | |
| | $\omega^2$ | $a_1$ | $\omega^2$ | $a_1$ | $\omega^2$ | $a_1$ | $\omega^2$ | $a_1$ | $\omega^2$ | $a_1$ | $\omega^2$ | $a_1$ | $\omega^2$ | $a_1$ | | |
| $A_1$ | 0.5937 | 0.333 | 0.9375 | 0.598 | 0.8452 | 0.540 | 0.8936 | 0.573 | 0.735 | 0.458 | 0.9783 | 0.627 | 0.6753 | 0.398 | | |
| $A_{1f}$ | 0.6356 | 0.371 | 0.8477 | 0.540 | 0.9658 | 0.621 | 0.7364 | 0.458 | 0.8456 | 0.540 | 0.9967 | 0.637 | 0.7986 | 0.504 | | |
| $A_2$ | 0.6953 | 0.424 | 0.7569 | 0.474 | 0.6973 | 0.424 | 0.7970 | 0.504 | 0.8478 | 0.547 | 0.8951 | 0.573 | - | - | | |
| $A_{2f}$ | 0.7354 | 0.458 | 0.8757 | 0.567 | 0.7072 | 0.441 | 0.9651 | 0.598 | 0.9996 | 0.643 | 1.0651 | 0.668 | - | - | | |
| $\beta \cdot 10^{-3}$ | 0.4652 | 0.202 | 0.6759 | 0.407 | 0.5671 | 0.313 | - | - | - | - | - | - | - | - | | |
| $c \cdot 10^{-4}$ | 1.1456 | 0.643 | 0.9954 | 0.637 | 1.2345 | 0.740 | - | - | - | - | - | - | - | - | | |
| $\overline{S}_T$ | 0.9983 | 0.637 | 0.8751 | 0.560 | 1.0784 | 0.677 | 0.7954 | 0.504 | 0.9672 | 0.615 | 0.8756 | 0.560 | 0.8971 | 0.580 | | |
| $\propto$ | - | - | - | - | - | - | - | - | - | - | - | - | 0.6978 | 0.424 | | |

In the analysis of cyclic deformation characteristics, the level of significance $\alpha_1 = 0.1$ has been accepted. The results of the reports confirm the inequality of $a_1 < 1 - \alpha_1 = 0.9$, indicating that the experimental data correspond to the theoretical law of normal distribution. Comparison of the calculation results of the criterion $\omega^2$ and the shape of the histograms of cyclic deformation parameters for all materials has suggested that the values of function $a_1$ correspond to the histograms similarly to the normal distribution law. This is apparently due to the greater sensitivity of the cyclic deformation curves, compared to monotonous tensile curves, chemical composition, and conditions of heat treatment of the material, as well as to the conditions of a more complex cyclic experiment.

The calculation was performed using statistical characteristics (arithmetic mean $\overline{x}$, standard deviation $s$, dispersion $D$, skewness $Sk$, and coefficient of variation $V$) for the three distribution laws: normal, log-normal, and Weibull law. The results of the calculations are given in Table A1. From the analysis of statistical characteristics, it can be established that it is more appropriate to describe the cyclic deformation parameters $A_1$, $A_{1f}$, $A_2$, $A_{2f}$, $\beta$, $c$, $\overline{S}_T$ and $\alpha$ as normal, and parameters $\beta$ and $c$ as log-normal distribution laws. When using these laws to describe cyclic parameters, the coefficient of variation takes the minimum values. The statistical characteristics of the Weibull law are little influenced by the characteristics of the normal distribution. Interestingly, the coefficient of variation of the normal distribution of deformation parameters $A_1$, $A_{1f}$, $A_2$, $A_{2f}$ clearly depends on the level of loading and decreases with the loading level increasing. The same trend is observed when histograms of the same parameters are considered. This is apparently due to the lower stability of the processes of cyclic hardening, and, in particular, the softening of the stages of fatigue and quasi-static damage.

Based on the chosen distribution law, the possible limits of the confidence intervals have been calculated at a designated confidence level 0.01 and 0.99, as well as for cyclic loading curves parameters. It is noteworthy that the limits of the confidence intervals of the parameters of cyclic deformation, as compared to the mechanical characteristics, also occupy a rather narrow band. The results of the calculations are presented in Table A2.

Figure 5 provides a comparison of the variation coefficients of the three distribution laws. From the analysis it also follows that, to describe the parameters of cyclic deformation $A_1$, $A_{1f}$, $A_2$, $A_{2f}$, $\beta$, $c$, $\overline{S}_T$ and $\alpha$ it is more appropriate to use the normal distribution law, for $\beta$ and $c$ using log-normal is more appropriate.

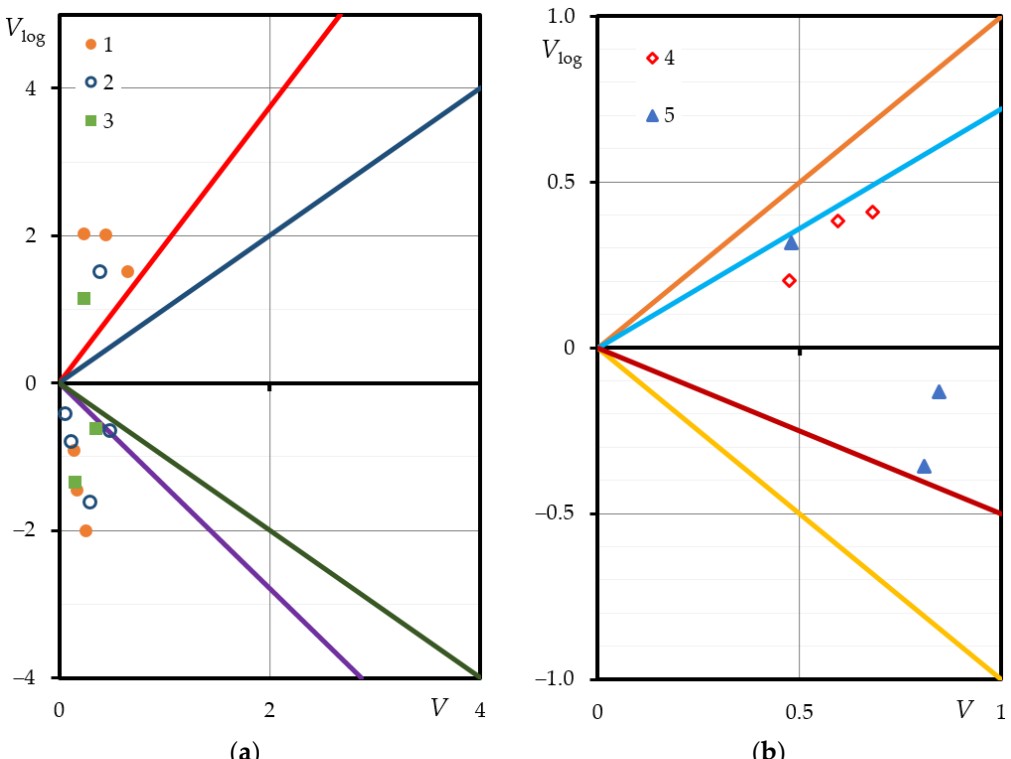

**Figure 5.** Coefficients of variation of cyclic properties (*x*-axis—normal and Weibull; *y*-axis—log-normal); (**a**) $1A_1, 2A_{1f}, 3A_2$; (**b**) $4\beta, 5c$.

The distributions of the cyclic deformation parameters investigated are shown in the probabilistic grid in Figure 6.

The deformation parameters of the low-cycle fatigue diagrams shown in the probability grid of the normal distribution confirm that these parameters are consistent with this law.

To estimate the variation of the parameters of cyclic deformation $A_1$, $A_{1f}$, $A_2$, $A_{2f}$, $\overline{S}_T$ and $\alpha$, ratio $K$ of the maximum values of statistical series of the cyclic parameter with its minimum value has been determined. The results of the calculation of relationship $K$ for the materials at the coordinates $s - K$, $V - K$ are shown in Figure 7. The resulting curves for parameters $A_1$, $A_{1f}$, $A_2$, $A_{2f}$, $\overline{S}_T$ and $\alpha$ (Figure 7a,b) indicate the following mathematical relationships between:

$$s^2 = 0.05(K - 1). \tag{24}$$

$$V^2 = 0.0884(K - 1). \tag{25}$$

Subjected to the condition: $s \geq 0$ and $V \geq 0$.

The resulting curves for parameters $\beta$ and $c$ (Figure 7c,d) designate the following mathematical relationships between:

$$s^2 = 0.046(K - 1). \tag{26}$$

$$V^2 = 0.09(K - 1). \tag{27}$$

Subjected to the condition: $s \geq 0$ and $-\infty \leq V \leq \infty$.

Equations (23)–(26) allow preliminary to estimate $s$, $V$ and $\bar{x}$ values of the cyclic deformation parameters.

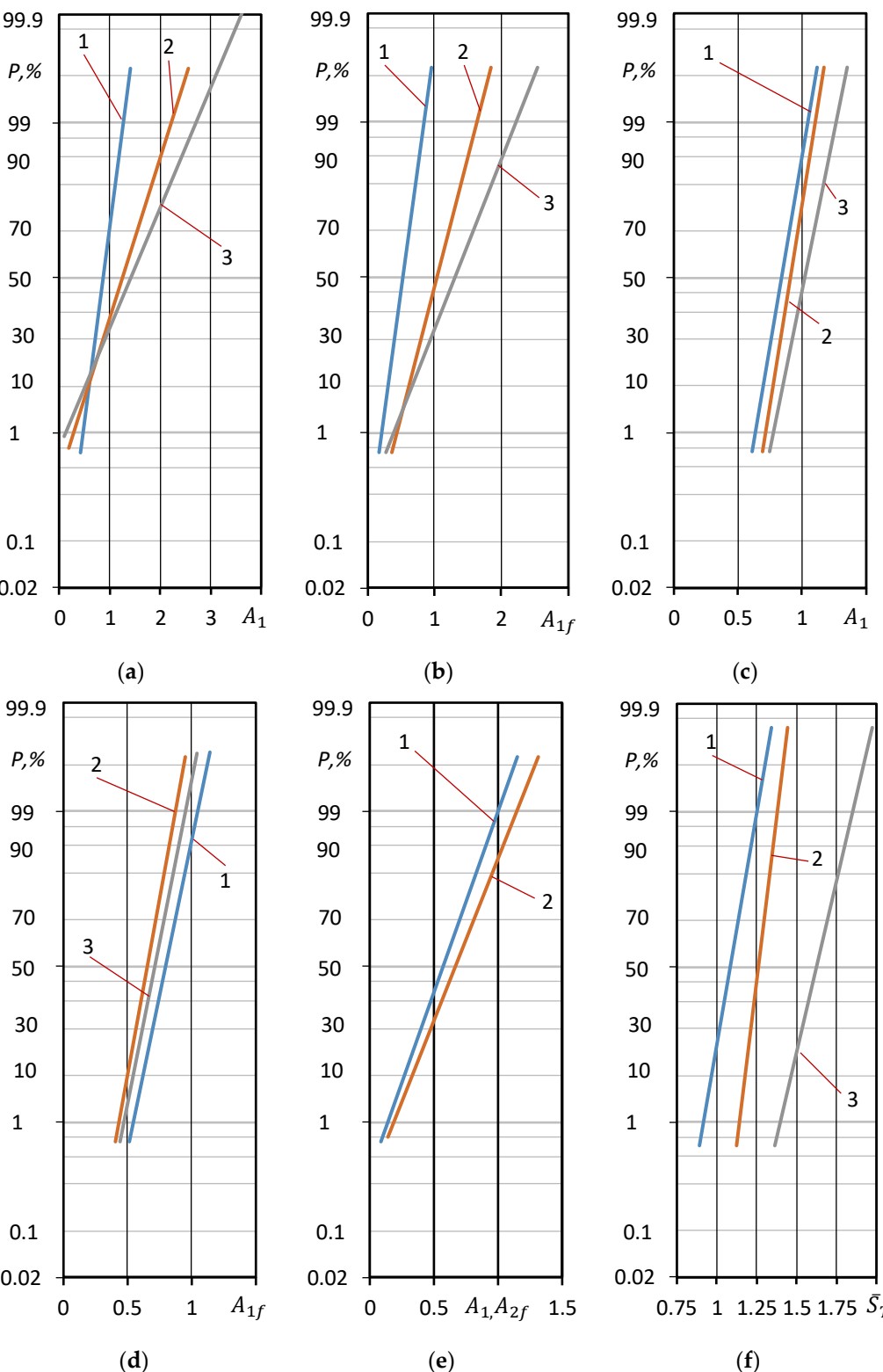

**Figure 6.** Log-normal distribution curves of the cyclic properties: (**a**,**b**)—steel 15Cr2MoVa ($1\bar{\sigma}_0 = 1.25$, $2\bar{\sigma}_0 = 1.12$, $3\bar{\sigma}_0 = 1.00$); (**c**,**d**)—steel C45 ($1\bar{\sigma}_0 = 1.50$, $2\bar{\sigma}_0 = 1.12$, $3\bar{\sigma}_0 = 1.00$); (**e**)—aluminum alloy D16T1 ($\bar{\sigma}_0 = 1.12$) (**f**)—normalized stress of cyclic proportionality limit: 1—steel 15Cr2MoVa, 2—steel C45, 3—aluminum alloy D16T.

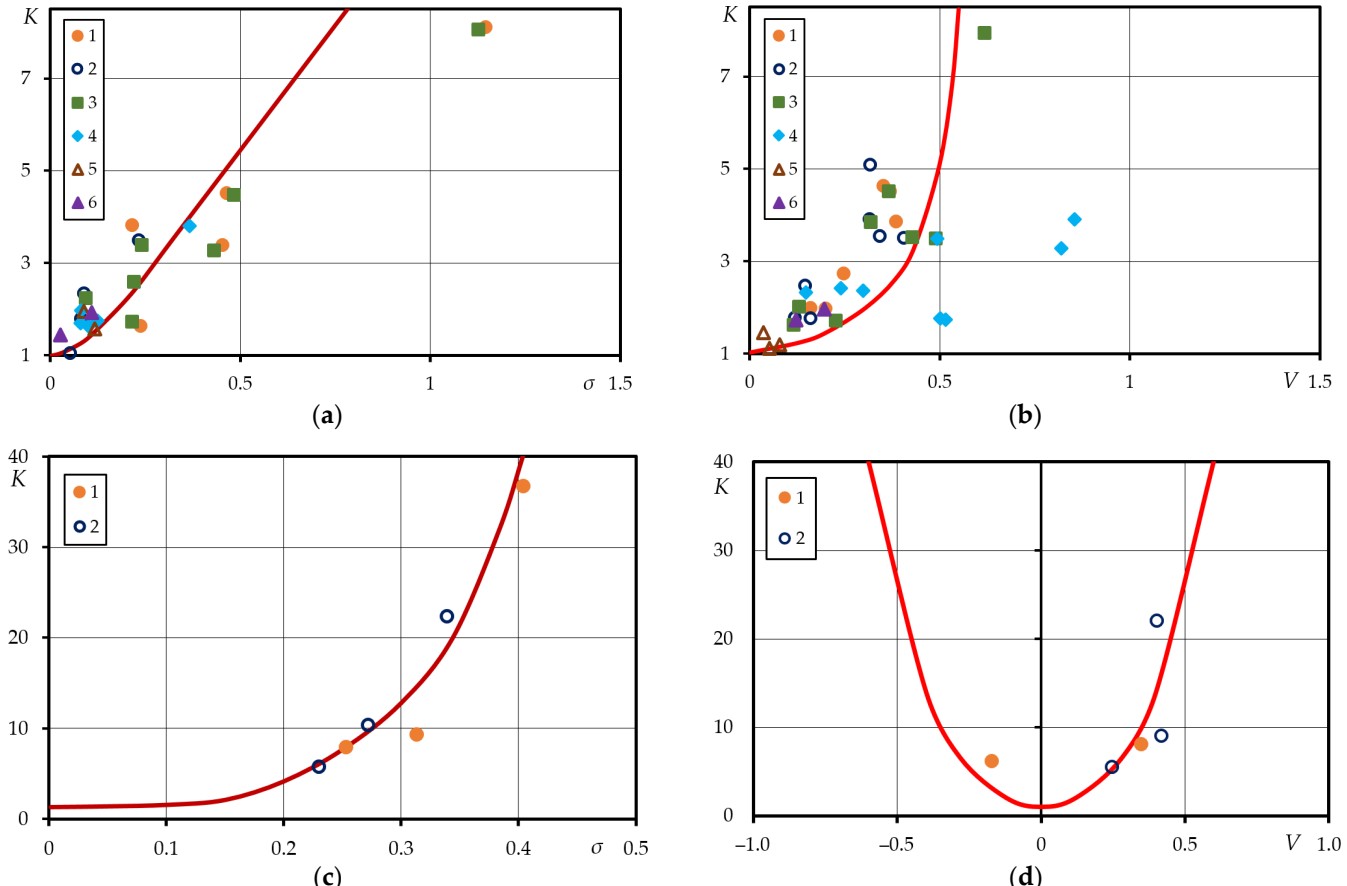

**Figure 7.** Dependence of relative value $K$ under the log-normal distribution law on mean square deviation $s$ and coefficient of variation $V$: (**a**,**b**) $1A_1$, $2A_{1f}$, $3A_2$, $4A_{2f}$, $5\overline{S}_T$, $6 \propto$; (**c**,**d**) $1\beta$, $2c$.

### 3.2. Probability Evaluation of Low-Cycle Fatigue Curves

For statistically investigated characteristics of the low cycle fatigue diagrams, probabilistic values have been established, allowing theoretical low-cycle fatigue curves to be estimated and assessed probabilistically. The specimen under stress-controlled loading fails due to quasi-static damage $d_k$ resulting from accumulated unilateral plastic deformation and fatigue damage $d_N$ resulting from cyclic plastic deformation, characterized by the width of the hysteresis loop $\overline{\delta}_k$:

$$d = d_K^q + d_N^l. \tag{28}$$

where fatigue damage:

$$d_N = \frac{\sum\limits_1^k \overline{\delta}_k}{\sum\limits_1^{k_N} \overline{\delta}_k}. \tag{29}$$

where $\sum\limits_1^k \overline{\delta}_k$—fatigue damage accumulated over $k$ half-cycles, o $\sum\limits_1^{k_N} \overline{\delta}_k$—fatigue damage accumulated before failure. Quasi-static damage:

$$d_K = \frac{\overline{e}_{pk}}{\overline{e}_u}. \tag{30}$$

To make the calculations more straightforward, the following was adopted: $q = l = 1$, therefore:

$$d = d_K + d_N = 1. \tag{31}$$

Under strain-controlled loading ($\varepsilon = constant$) strain is constrained, thus, no unilateral strain is accumulated and there is no quasi-static damage.

The fatigue curve at coordinates $lg\bar{\delta} - lgk_c$ is a straight line. Using the equation of the line [36,37] one obtains the following.

$$lg\bar{\delta} = -m \cdot lgk_c + lgC \text{ or } \bar{\delta}k_c^m = C. \tag{32}$$

Using Equations (28)–(32), the damage condition can be written:

$$\left( \frac{\sum\limits_{1}^{k_c} \bar{\delta}_k}{\sum\limits_{1}^{k_N} \bar{\delta}_k} \right)^q + \left( \frac{\bar{e}_{pk_C}}{\bar{e}_u} \right)^l = 1. \tag{33}$$

In the formulation of the theoretical curves for the low-cycle fatigue, only the fatigue damage is considered, and stress-controlled loading is treated as a nonstationary strain-controlled loading, with the cumulative damage of one half-cycle expressed by the equation:

$$d_N = \frac{\bar{\delta}_k}{\sum\limits_{1}^{k_c} \bar{\delta}_k}. \tag{34}$$

In this case, the condition for the crack initiation is:

$$\frac{\bar{\delta}_1}{\sum\limits_{1}^{k_{c1}} \bar{\delta}_k} + \frac{\bar{\delta}_2}{\sum\limits_{1}^{k_{c2}} \bar{\delta}_k} + \frac{\bar{\delta}_3}{\sum\limits_{1}^{k_{c3}} \bar{\delta}_k} + \ldots + \frac{\bar{\delta}_c}{\sum\limits_{1}^{k_{ck_c}} \bar{\delta}_k} = 1. \tag{35}$$

where $\sum\limits_{1}^{k_{c1}} \bar{\delta}_k$ accumulated fatigue damage prior to failure with hysteresis loop width $\bar{\delta}_1$; $\sum\limits_{1}^{k_{c2}} \bar{\delta}_k$ accumulated fatigue damage up to the crack initiation $\bar{\delta}_2$, etc. The fatigue curve under strain-controlled loading ($e = constant$) in coordinates $lg\bar{\delta} - lgk_c$ provides:

$$\bar{\delta} = C_2 k_c^{-m_2} \text{ or } \sum\limits_{1}^{k_c} \bar{\delta}_k = C_2 k_c^{1-m_2}. \tag{36}$$

By using the coordinates $lg\bar{\varepsilon} - lgk_c$, one obtains $\bar{\varepsilon}k_c^{m_1} = C_3$ and:

$$k_c = \frac{C_3^{1/m_1}}{\bar{\varepsilon}^{1/m_1}}. \tag{37}$$

By writing Equation (37) to Equation (36) gives:

$$\sum\limits_{1}^{k_c} \bar{\delta}_k = C_2 \frac{C_3^{\frac{1-m_2}{m_1}}}{\bar{\varepsilon}^{\frac{1-m_2}{m_1}}}. \tag{38}$$

After writing $(1 - m_2)/m_1$ and inserted into Equation (35) one obtains:

$$\frac{\bar{\delta}_1\bar{\varepsilon}_1^{m_3}}{C_2 C_3^{m_3}} + \frac{\bar{\delta}_2\bar{\varepsilon}_2^{m_3}}{C_2 C_3^{m_3}} + \frac{\bar{\delta}_3\bar{\varepsilon}_3^{m_3}}{C_2 C_3^{m_3}} + \ldots + \frac{\bar{\delta}_{k_c}\bar{\varepsilon}_{k_c}^{m_3}}{C_2 C_3^{m_3}} = 1. \tag{39}$$

Under stress-controlled loading for cyclically stable material:

$$\frac{\overline{\delta}_k}{\sum\limits_1^{k_c} \overline{\delta}_k} = \frac{k}{k_c}. \tag{40}$$

Therefore, the summation of the comparable cyclic deformations may be substituted by the summation of the comparable lifetimes:

$$\sum_i \frac{k_i}{k_{c_i}} = 1. \tag{41}$$

In this case $\overline{\delta}_k = \overline{\delta}_{vid} = constant$, using Equations (36) and (41), one obtains:

$$\sum_i k_i \overline{\delta}_k^{-1/m_2} = C_2^{1/m_2} \text{ and}$$

$$\left[ \sum_i k_i \overline{\delta}_k^{-1/m_2} = C_2^{1/m_2} \right]^{m_2} = \left( \overline{\delta}_1^{-1/m_2} + \overline{\delta}_2^{-1/m_2} + \overline{\delta}_3^{-1/m_2} + \dots \overline{\delta}_k^{-1/m_2} \right)^{m_2} = C_2. \tag{42}$$

Using Equation (36), it is achievable to calculate the fatigue damage of cyclically stable, cyclically softening, and cyclically hardening materials. A piecewise approximation of the generalized strain diagram produces Equation (36) as follows:

$$\overline{\sigma}_0 = \left\{ \frac{C_2}{\left[ \left( A_1^{1/m_2} + A_1^{1/m_2} \right) N \right]^{m_2}} - \frac{\overline{S}_T}{2} \right\}^{m_0} \tag{43}$$

For the determination of the probabilistic curves, the values of the probabilistic coefficients $A_1$, $A_2$, $\overline{S}_T$, $m_0$, $c \cdot 10^{-4}$ were used (Table 4). The comparison of the calculation and experimental data of these graphs is shown in Figure 8a,b, respectively.

**Table 4.** Probabilistic values of the parameters for resistance to cyclic deformation.

| Parameters | Probability, % | | | | | | |
|:---:|:---:|:---:|:---:|:---:|:---:|:---:|:---:|
| | 1 | 10 | 30 | 50 | 70 | 90 | 99 |
| | | | 15Cr2MoVa | | | | |
| $A_1$ | 0.23 | 0.87 | 1.32 | 1.60 | 1.94 | 2.40 | 3.04 |
| $A_2$ | 0.30 | 0.94 | 1.34 | 1.64 | 2.00 | 2.43 | 3.06 |
| $\overline{S}_T$ | 1.15 | 1.20 | 1.25 | 1.28 | 1.30 | 1.35 | 1.40 |
| $m_0$ | 0.15 | 0.17 | 0.19 | 0.21 | 0.23 | 0.26 | 0.30 |
| $c \cdot 10^{-4}$ | 1.1 | 3.0 | 4.9 | 7.4 | 11.0 | 21.0 | 50.0 |
| | | | C 45 | | | | |
| $A_1$ | 0.60 | 0.76 | 087 | 0.95 | 1.04 | 1.15 | 1.29 |
| $A_2$ | 0.61 | 0.80 | 0.89 | 0.96 | 1.05 | 1.16 | 1.30 |
| $\overline{S}_T$ | 0.92 | 1.02 | 1.07 | 1.11 | 1.15 | 1.21 | 1.29 |
| $m_0$ | 0.14 | 0.16 | 0.18 | 0.19 | 0.20 | 0.22 | 0.25 |

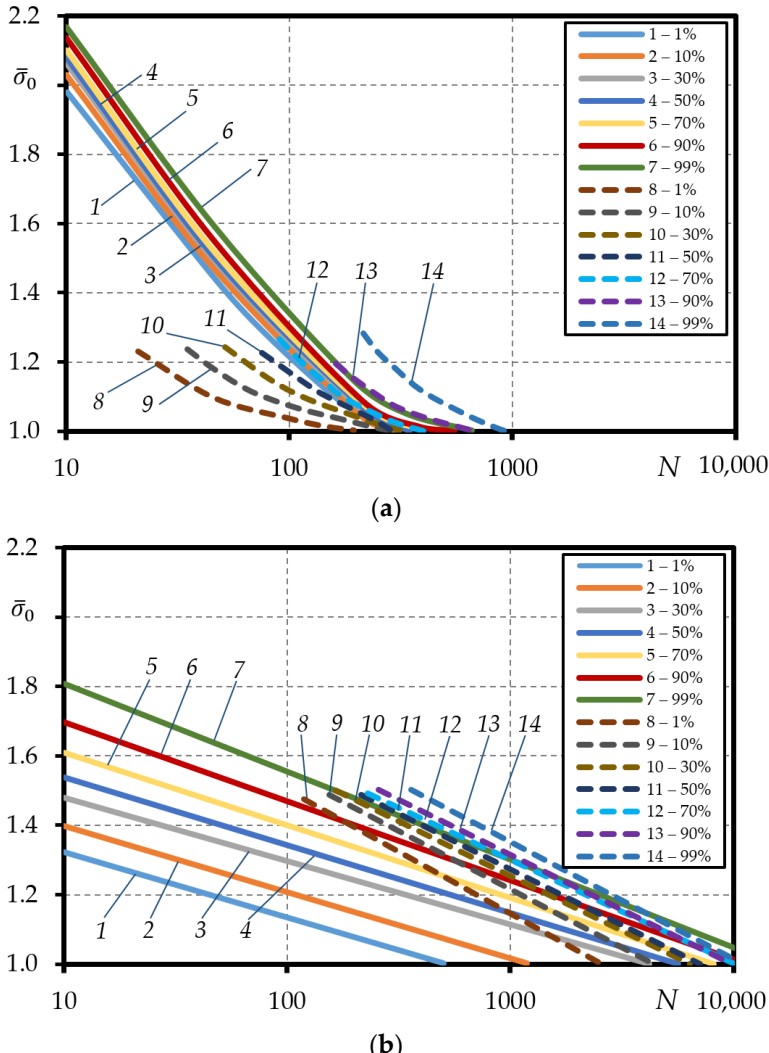

**Figure 8.** Comparison of the experimental (dashed lines) and theoretical (straight lines) curves under loading with controlled stress (1–7 analytical probability 1–99%; 8–14 experimental probability 1–99%): (**a**)—steel 15Cr2MoVa (Equation (11)); (**b**)—steel C45.

Using Equation (37) for calculation methodology, it was obtained that the low-cycle curves reach a suitable order, that is, the curves for steel 15Cr2MoVa and steel C45 of 1% show the lowest lifetime, while 99% the highest. Analytically determined curves for steel 15Cr2MoVa produce a relatively narrow range, whose magnitude is dependent on the loading level $\overline{\sigma}_0$. From the generated graphs, it was determined that for a curve of 1% curve at a loading level $\overline{\sigma}_0$, the durability $N = 10$, and for a curve of 99%, the durability $N = 2$; at a loading level $\overline{\sigma}_0 = 1.6$ for a 1% curve, durability $N = 48$ and for a curve of 99%, the durability $N = 82$; and at a loading level $\overline{\sigma}_0 = 1.2$ for a 1% curve, durability $N = 380$ and for a curve of 99%, the durability $N = 570$. The experimentally determined durability curves for steel 15Cr2MoVa are shown in Figure 8a. It can be observed that the distribution and scatter of the curves are significantly higher compared to the analytically calculated ones. The analytical durability curves have a distribution between 70% and 90% of the probabilistic experimental curves at a loading level of $\overline{\sigma}_0 = 1.0$ and between 15% and 50% of the probabilistic curves at a loading level of $\overline{\sigma}_0 = 1.0$.

The comparison of the experimental and theoretical curves of steel C45 under controlled strain loading is shown in Figure 8b. Probability curves were calculated analytically using Equation (37) and the parameters in Table 4. The probabilistic values of the constants $C_2$ and $m$ were derived from the probability curves $lg\overline{\delta}_1 - lgk_c$ for strain-controlled loading

and are presented in Table 4. Hysteresis loop width $\overline{\delta}_1$ was determined using Equation (30). It can be seen that the probability curves reach a suitable order, i.e., the 1% curve is the lowest and the 99% curve the highest. It can be seen that there is a large scatter (Figure 8b), which indicates a high dependence of durability $N$ on the loading level $\overline{\sigma}_0$. The range of curves becomes narrower with increasing durability. When comparing the experimental results for steel 15Cr2MoVa and C45 with the analytical results, the opposite effect can be observed: the experimental curves have a much tighter range and scatter than the analytically estimated ones. At loading level $\overline{\sigma}_0 = 1.0$ the experimentally estimated durability of 1% to 99% is distributed between the analytical curves of 15% and 90%.

## 4. Conclusions

Based on the results, we make the following observations:

1. Statistical investigations of the elastic-plastic parameters of the low-cycle fatigue curves showed that these parameters vary more than the statistical parameters of the mechanical characteristics. This is probably due to the sensitivity of the cyclic strain characteristic to variations in chemical composition and thermal and technological processing conditions, as well as to the effect of the considerable relative experimental error on the hysteresis loop width. As the low cycle load level decreases, the scatter of low cycle deformation characteristics increases. This is apparently due to the lower stability of the hardening and softening processes in the fatigue and transient failure zones compared to the quasi-static failure zone.

2. It has been established that relative coordinates $\overline{\sigma} - \overline{\varepsilon}$ should be used under stress-controlled loading. The use of absolute values, due to the significant dispersion of the proportionality limit $e_{pr}$, leads to a large dispersion of the deformation and failure parameters of the low-cycle deformation.

3. Analysis of the histograms shows that for all materials and loading levels investigated, there is a positive asymmetry, but as the sampling rate increases, the histograms become smoother and approximate to a normal distribution law.

4. Calculation of cyclic characteristics for all materials using mean absolute deviation, variation amplitude, and compatibility criteria has shown that the results could be represented by the normal distribution law.

5. The scattering of the parameters of cyclic elastic-plastic deformation curves is satisfactorily described by the laws of normal and log-normal distribution, with obvious improvement in fit with the increasing sample size, or when tested material groups are grouped by chemical composition, surface hardening, and heat treatment.

6. The low-cycle deformation diagram becomes flatter as the load level increases, so that small stress variations lead to larger hysteresis loop width variations. This increases the dispersion of durability.

7. The analysis performed on the low-cycle fatigue curves under controlled stress loading shows only fatigue damage. For anisotropic materials steel 15Cr2MoVa and steel C45, satisfactory agreement between the experimental and analytical low cycle fatigue curves is only possible at a loading level $\overline{\sigma}_0 = 1.0$ and $\sigma_{pr}$. The influence of damage increases at higher values of the quasi-static load level.

**Author Contributions:** Conceptualization, Ž.B., V.L.; methodology, Ž.B., V.L.; software, V.L.; validation, Ž.B., V.L.; formal analysis, Ž.B.; investigation, Ž.B., V.L.; resources, Ž.B.; data curation, Ž.B., V.L.; writing—original draft preparation, Ž.B., V.L.; writing—review and editing, V.L.; visualization Ž.B., V.L.; supervision, Ž.B., V.L.; project administration, Ž.B. All authors have read and agreed to the published version of the manuscript.

**Funding:** This research received no external funding.

**Institutional Review Board Statement:** Not applicable.

**Informed Consent Statement:** Not applicable.

**Data Availability Statement:** Not applicable.

**Acknowledgments:** All co-authors have contributed equally. The authors declare that they have no known competing financial interests or personal relationships that could have appeared to influence the work reported in this paper.

**Conflicts of Interest:** The authors declare no conflict of interest.

## Nomenclature

| | |
|---|---|
| $a_1$ | tabular function value |
| $A_1$, $A_2$ | constant describing the first and second semi-cycle form respectively |
| $A_{1f}$, $A_{2f}$ | fictitious constants of the first and second semi-cycle form respectively |
| $c$ | materials hardening or softening equation coefficient |
| $C$, $m$ | constants of the Coffin–Manson equation |
| $C_1$, $C_2$, $C_3$ | constants of the fatigue curve under strain-controlled loading |
| $d_k$, $d_N$ | quasi-static and fatigue damage respectively |
| $D$ | dispersion |
| $e$ | monotonous strain (%) |
| $e_0$ | strain if initial (0 semi-cycle) loading (%) |
| $\bar{e}_0$ | strain if initial (0 semi-cycle) loading normalized to proportional limit strain (%) |
| $e_f$ | fracture strain (%) |
| $e_k$ | cyclic strain of $k$ semi-cycle (%) |
| $e_{pk}$ | accumulated plastic strain after loading semi-cycles in the direction of tension (%) |
| $e_{pr}$ | proportional limit strain (%) |
| $e_u$ | static loading monotonous strain (%) |
| $E$ | modulus of elasticity (MPa) |
| $k$ | number of loading semi-cycle under controlled stress |
| $k_c$ | number of loading semi-cycle under controlled strain |
| $m_1$, $m_2$, $m_3$, | constants of the fatigue curve under strain-controlled loading |
| $P$ | probability |
| $q$, $l$ | equation exponent |
| $s$ | standard deviation |
| $S$ | cyclic stress at cyclic loading (MPa) |
| $Sk$ | skewness |
| $S_k$ | cyclic stress of $k$ semi-cycle (MPa) |
| $\bar{S}_k$ | cyclic stress of $k$ semi-cycles normalized respectively to proportional limit stress (MPa) |
| $S_T$ | stress of cyclic proportionality limit (MPa) |
| $\bar{S}_T$, $\bar{S}_{Tk}$ | stress normalized respectively to proportional limit stress (MPa) |
| $\bar{x}$ | sample mean |
| $x_p^U$ | the upper endpoint of the confidence interval |
| $x_p^L$ | the lower endpoint of the confidence intervals |
| $V$ | coefficient of variation |
| *Greek symbols* | |
| $\alpha$, $\beta$ | constants characterizing materials hardening or softening |
| $\alpha_1$ | level of significance |
| $\delta_1$ | width of hysteresis loop for first semi-cycle (%) |
| $\delta_{1f}$, $\delta_{2f}$ | fictitious width of hysteresis loop of the first and second semi-cycle respectively (%) |
| $\delta_k$ | width of hysteresis loop for $k$ semi-cycle (%) |
| $\delta_{k4}$, $\delta_{k4f}$, | width of hysteresis loop for fourth semi-cycle (%) |
| $\bar{\delta}$ | average width of hysteresis loop (%) |
| $\bar{\delta}_k$ | width of hysteresis loop normalized to proportional limit strain |
| $\psi$ | percent area reduction (%) |
| $\psi_u$ | percent area reduction at failure (%) |
| $\varepsilon$ | cyclic strain at cyclic loading (%) |
| $\varepsilon_k$ | cyclic strain of $k$ semi-cycle (%) |
| $\bar{\varepsilon}_k$ | cyclic strain of $k$ semi-cycles normalized respectively to proportional limit strain (%) |
| $\bar{\varepsilon}_{pk}$ | cyclic accumulated plastic strain normalized respectively to proportional limit strain (%) |

| | |
|---|---|
| $\sigma$ | monotonous stress (MPa) |
| $\sigma_0$ | stress of initial (0 semi-cycle) loading (MPa) |
| $\overline{\sigma}_0$ | stress of initial (0 semi-cycle) loading normalized to proportional limit stress (MPa) |
| $\sigma_{0.2}$ | elastic limit or yield strength, the stress at which 0.2% plastic strain occurs (MPa) |
| $\sigma_f$ | fracture stress (MPa) |
| $\sigma_k$ | cyclic stress of $k$ semi-cycle (MPa) |
| $\sigma_{pr}$ | proportional limit stress (MPa); |
| $\sigma_u$ | ultimate tensile stress (MPa) |
| $\overline{\sigma}$ | normalized to proportional limit cyclic stress (MPa) |
| $\omega^2$ | Smirnov compatibility criterion |
| $W_a^2$ | critical value of Smirnov criterion ($W_{0.1}^2 = 0.104$; $W_{0.2}^2 = 0.126$; $W_{0.01}^2 = 0.178$) |
| $\Phi(\hat{z}_i)$ | Laplace function value |

## Appendix A

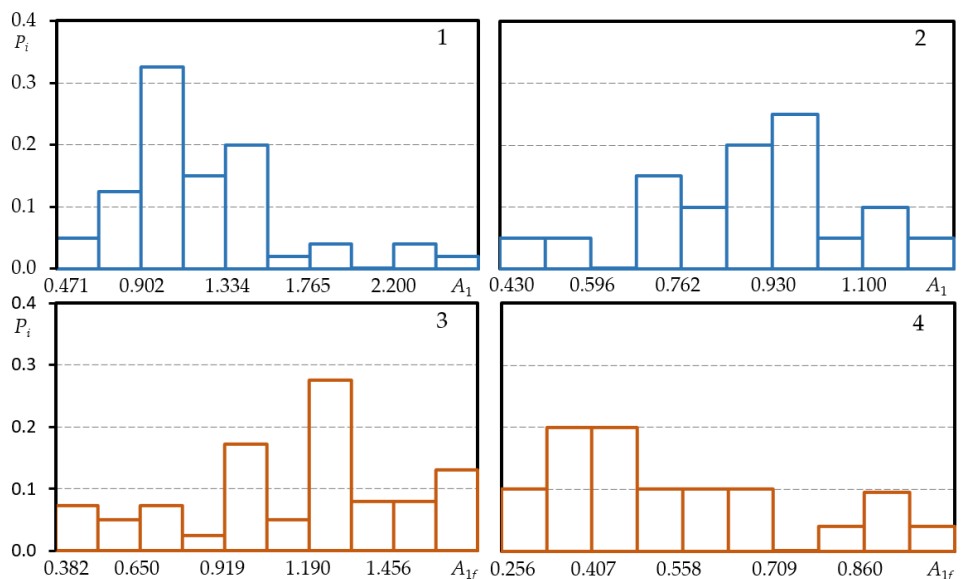

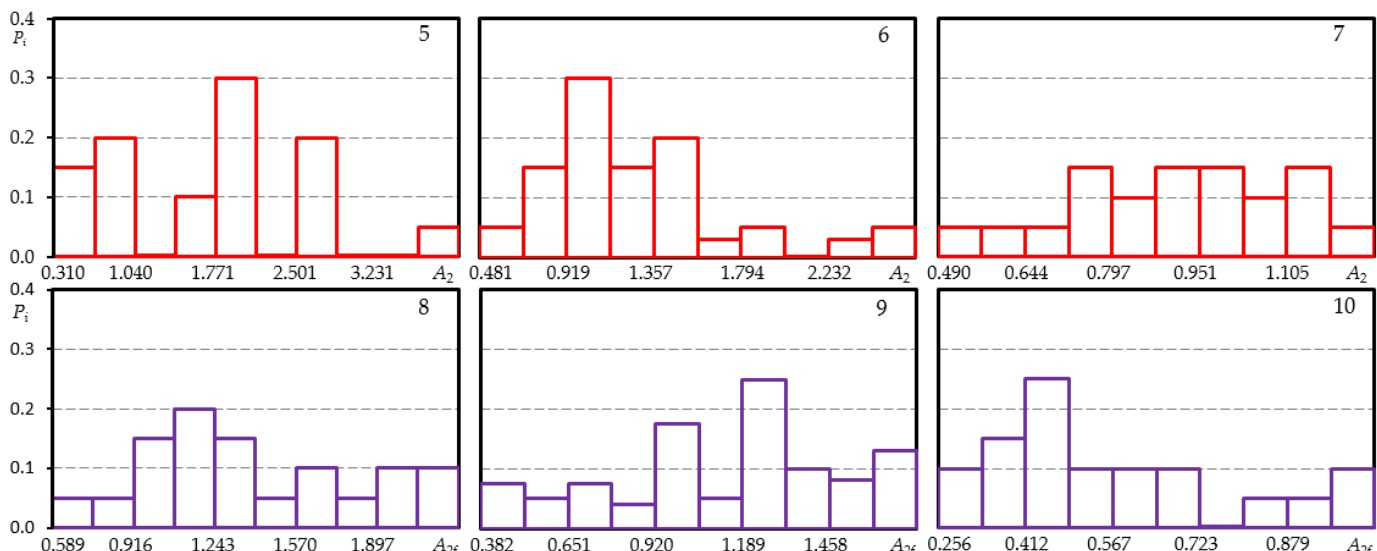

**Figure A1.** Histograms of low cyclic load diagram characteristics of steel 15Cr2MoVa; loading level $\overline{\sigma}_0 = 1.00$—5, 8; loading level $\overline{\sigma}_0 = 1.12$—1, 3, 6, 9; loading level $\overline{\sigma}_0 = 1.25$—2, 4, 7, 10 (1, 2—$A_1$; 3, 4—$A_{1f}$; 5–7—$A_2$; 8–10—$A_{2f}$).

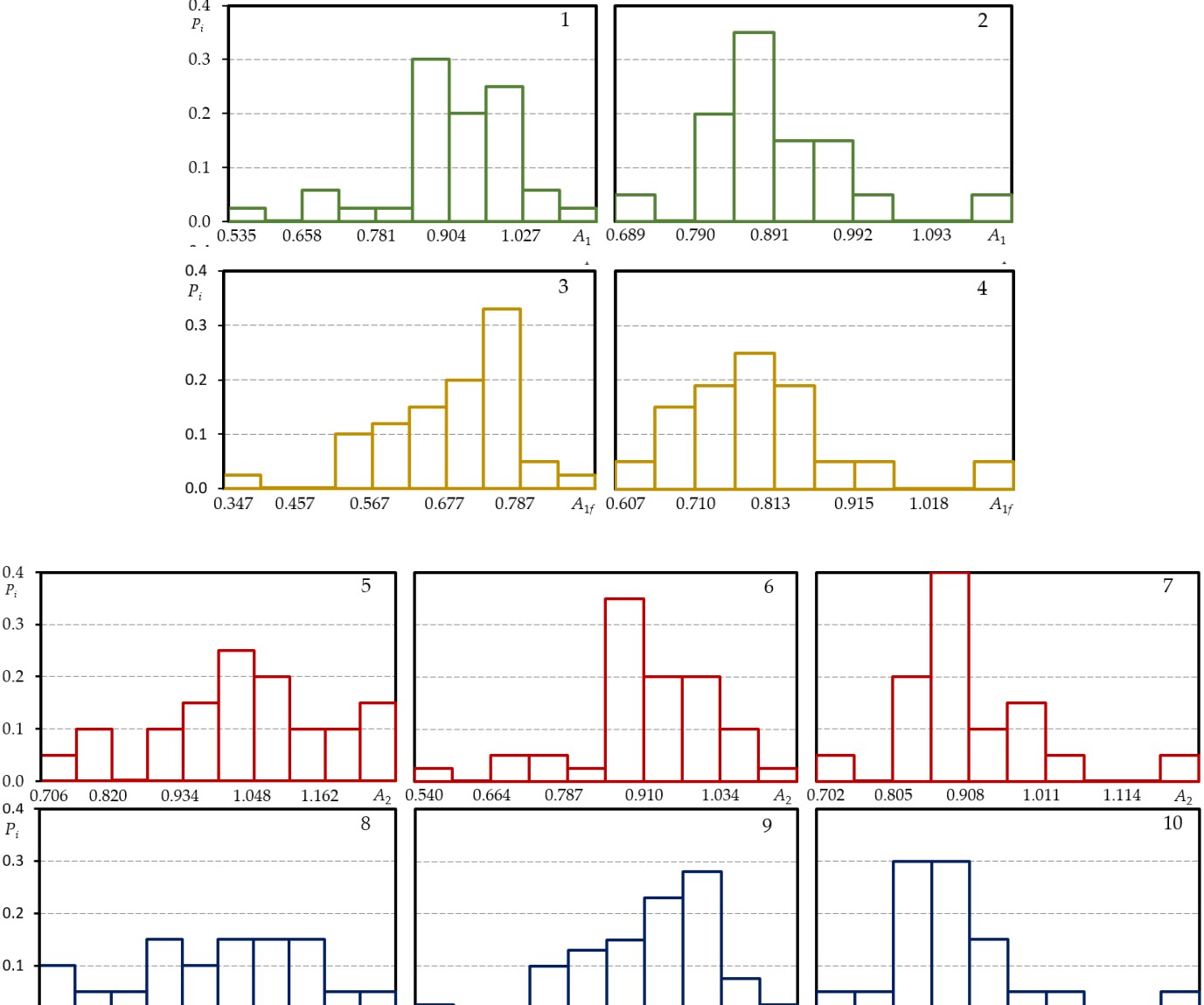

**Figure A2.** Histograms of low cyclic load diagram characteristics of steel C45; loading level $\overline{\sigma}_0 = 1.00$—5, 8; loading level $\overline{\sigma}_0 = 1.12$—1, 3, 6, 9; loading level $\overline{\sigma}_0 = 1.25$—2, 4, 7, 10 (1, 2—$A_1$; 3, 4—$A_{1f}$; 5–7—$A_2$; 8–10—$A_{2f}$).

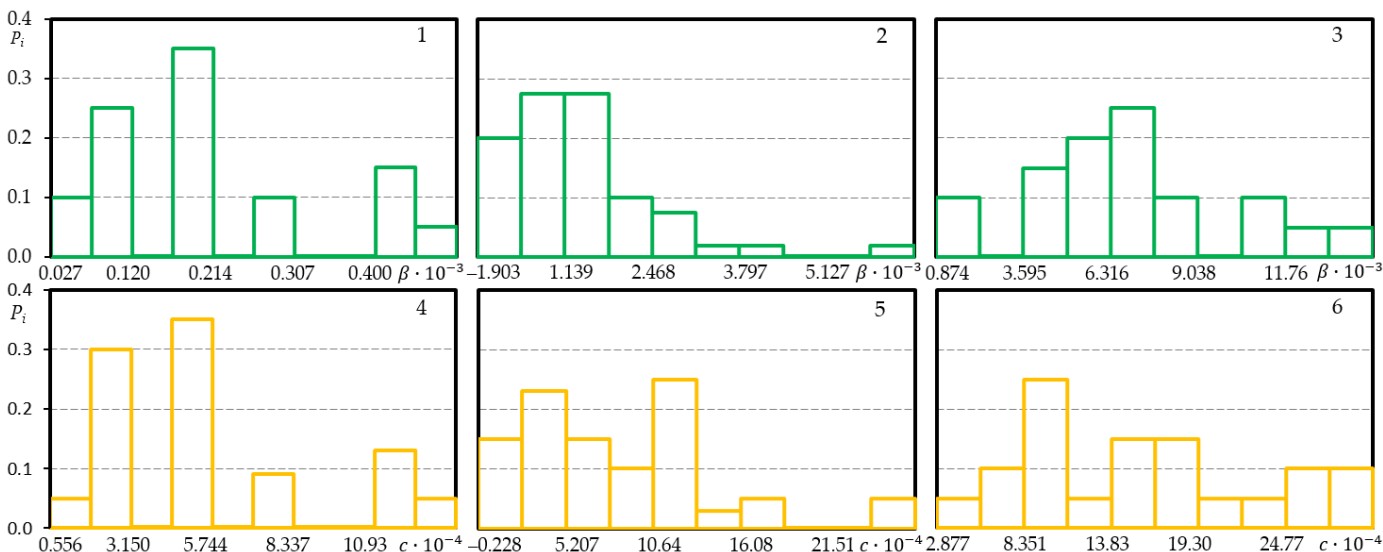

**Figure A3.** Histograms of low cyclic load diagram characteristics of steel 15Cr2MoVa C45; loading level $\overline{\sigma}_0 = 1.00$—1, 4; loading level $\overline{\sigma}_0 = 1.12$ 2, 5; loading level $\overline{\sigma}_0 = 1.25$ 3, 6.

## Appendix B

The calculation has been made using statistical characteristics (arithmetic mean $\overline{x}$, standard deviation $s$, dispersion $D$, skewness $Sk$, and coefficient of variation $V$) for the three distribution laws: normal, log-normal, and Weibull law. The results of the calculations are given in Table A1.

**Table A1.** Statistical properties for normal, log-normal, and Weibull distribution of cyclic properties.

| Cyclic Properties | Material | $\overline{\sigma}_0$ | $n$ | Normal | | | | | Log-Normal | | | | | Weibull | | | |
|---|---|---|---|---|---|---|---|---|---|---|---|---|---|---|---|---|---|
| | | | | $\overline{x}$ | $s$ | $D$ | $Sk$ | $V$ | $\overline{x}$ | $s$ | $D$ | $Sk$ | $V$ | $\overline{x}$ | $s$ | $Sk$ | $V$ |
| $A_1$ | 15Cr2MoVa | 1.00 | 20 | 1.8175 | 1.1206 | 1.2558 | 1.329 | 0.616 | 1.5293 | 0.2676 | 0.0716 | −0.135 | 1.451 | 1.8067 | 1.1206 | 1.555 | 0.620 |
| | | 1.12 | 40 | 1.2237 | 0.4515 | 0.2038 | 1.151 | 0.365 | 1.1636 | 0.1481 | 0.0219 | 0.284 | 2.251 | 1.1884 | 0.4514 | 1.243 | 0.379 |
| | | 1.25 | 20 | 0.8915 | 0.2146 | 0.0460 | 0.008 | 0.241 | 0.8649 | 0.1132 | 0.0128 | −0.713 | −1.796 | 0.8915 | 0.2145 | −0.009 | 0.241 |
| | C45 | 1.00 | 20 | 1.0477 | 0.2201 | 0.0484 | −0.470 | 0.210 | 1.0215 | 0.1063 | 0.0113 | −1.491 | 1.148 | 1.0477 | 0.2201 | −0.550 | 0.210 |
| | | 1.25 | 40 | 0.9135 | 0.1149 | 0.0132 | −0.965 | 0.126 | 0.9056 | 0.0598 | 0.0036 | −1.412 | −1.390 | 0.9135 | 0.1143 | −1.041 | 0.126 |
| | | 1.50 | 20 | 0.8865 | 0.0974 | 0.0095 | 1.004 | 0.109 | 0.8817 | 0.0459 | 0.0021 | 0.617 | −0.839 | 0.8503 | 0.0974 | 1.175 | 0.114 |
| | D16T1 | 1.15 | 20 | 0.6092 | 0.2212 | 0.0489 | 0.475 | 0.363 | 0.5714 | 0.1618 | 0.0262 | −0.147 | −0.666 | 0.6092 | 0.2212 | 0.558 | 0.363 |
| $A_{1f}$ | 15Cr2MoVa | 1.00 | 20 | 1.3849 | 0.4455 | 0.1985 | −0.088 | 0.322 | 1.3034 | 0.1672 | 0.279 | −1.227 | 1.453 | 1.3849 | 0.4455 | −0.103 | 0.322 |
| | | 1.12 | 40 | 1.1342 | 0.3512 | 0.1234 | −0.325 | 0.309 | 1.0711 | 0.1578 | 0.0249 | −0.919 | 5.292 | 1.1342 | 0.3512 | −0.351 | 0.309 |
| | | 1.25 | 20 | 0.5466 | 0.2195 | 0.0482 | 0.424 | 0.402 | 0.5029 | 0.1897 | 0.0359 | −0.589 | −0.636 | 0.5466 | 0.2195 | 0.496 | 0.402 |
| | C45 | 1.00 | 20 | 0.7191 | 0.1047 | 0.0109 | −0.364 | 0.146 | 0.7114 | 0.0662 | 0.0044 | −0.577 | −0.448 | 0.7191 | 0.1047 | −0.426 | 0.146 |
| | | 1.25 | 40 | 0.6916 | 0.0957 | 0.0091 | −0.885 | 0.138 | 0.6842 | 0.0668 | 0.0045 | −1.546 | −0.405 | 0.6916 | 0.0957 | −0.956 | 0.138 |
| | | 1.50 | 20 | 0.7908 | 0.1015 | 0.0103 | 1.189 | 0.128 | 0.7851 | 0.0527 | 0.0028 | 0.771 | −0.502 | 0.7618 | 0.1015 | 1.391 | 0.133 |
| | D16T1 | 1.15 | 20 | 0.7281 | 0.2334 | 0.0545 | −0.021 | 0.320 | 0.6872 | 0.1604 | 0.0257 | −0.939 | −0.985 | 0.7281 | 0.2334 | −0.025 | 0.321 |
| $A_2$ | 15Cr2MoVa | 1.00 | 20 | 1.8269 | 1.1217 | 1.2583 | 1.319 | 0.614 | 1.5389 | 0.2668 | 0.0712 | −0.137 | 1.425 | 1.8203 | 1.1217 | 1.543 | 0.616 |
| | | 1.12 | 40 | 1.2532 | 0.4659 | 0.2121 | 1.163 | 0.367 | 1.1816 | 0.1483 | 0.0220 | 0.305 | 2.047 | 1.2086 | 0.4606 | 1.256 | 0.381 |
| | | 1.25 | 20 | 0.9176 | 0.2111 | 0.0445 | 0.028 | 0.230 | 0.8931 | 0.4064 | 0.0113 | −0.598 | −2.167 | 0.9176 | 0.2111 | 0.033 | 0.230 |
| | C45 | 1.00 | 20 | 1.0524 | 0.2196 | 0.0482 | −0.521 | 0.209 | 1.0263 | 0.1061 | 0.0113 | −1.534 | 9412 | 1.0524 | 0.2196 | −0.609 | 0.209 |
| | | 1.25 | 40 | 0.9211 | 0.1155 | 0.0133 | −0.954 | 0.125 | 0.9131 | 0.0596 | 0.0035 | −1.402 | −1.511 | 0.9210 | 0.1155 | −1.030 | 0.125 |
| | | 1.50 | 20 | 0.8987 | 0.0975 | 0.0095 | 1.115 | 0.108 | 0.8940 | 0.0451 | 0.0020 | 0.709 | −0.926 | 0.8560 | 0.0974 | 1.304 | 0.114 |
| $A_{2f}$ | 15Cr2MoVa | 1.00 | 20 | 1.3976 | 0.4440 | 0.1972 | −0.067 | 0.318 | 1.3186 | 0.1629 | 0.0265 | −1.159 | 1.357 | 1.3976 | 0.4440 | −0.078 | 0.318 |
| | | 1.12 | 40 | 1.1358 | 0.3519 | 0.1239 | −0.325 | 0.309 | 1.0725 | 0.1579 | 0.0249 | −0.920 | 5.192 | 1.1358 | 0.3519 | −0.351 | 0.309 |
| | | 1.25 | 20 | 0.5674 | 0.2302 | 0.0530 | 0.464 | 0.406 | 0.5216 | 0.1901 | 0.0361 | −0.555 | −0.673 | 0.5674 | 0.2302 | 0.542 | 0.406 |
| | C45 | 1.00 | 20 | 0.7287 | 0.1109 | 0.0123 | −0.188 | 0.152 | 0.7203 | 0.0684 | 0.0047 | −0.423 | −0.481 | 0.7287 | 0.1109 | −0.220 | 0.152 |
| | | 1.25 | 40 | 0.6995 | 0.0954 | 0.0091 | −0.912 | 0.136 | 0.6922 | 0.0658 | 0.0043 | −1.583 | −0.412 | 0.6995 | 0.0954 | −0.985 | 0.136 |
| | | 1.50 | 20 | 0.8069 | 0.0999 | 0.0099 | 1.333 | 0.124 | 0.8016 | 0.0506 | 0.0026 | 0.886 | −0.527 | 0.7587 | 0.0999 | 1.559 | 0.132 |

**Table A1.** *Cont.*

| Cyclic Properties | Material | $\bar{\sigma}_0$ | n | Normal | | | | | Log-Normal | | | | | Weibull | | | |
|---|---|---|---|---|---|---|---|---|---|---|---|---|---|---|---|---|---|
| | | | | $\bar{x}$ | s | D | Sk | V | $\bar{x}$ | s | D | Sk | V | $\bar{x}$ | s | Sk | V |
| $\beta \cdot 10^{-3}$ | 15Cr2MoVa | 1.00 | 20 | 0.2355 | 0.1915 | 0.0367 | 2.012 | 0.813 | 0.1840 | 0.3098 | 0.0959 | 0.121 | −0.421 | 0.2209 | 0.1915 | 2.354 | 0.866 |
| | | 1.12 | 40 | 1.4325 | 1.2270 | 1.5056 | 1.679 | 0.856 | 0.9942 | 0.4037 | 0.1629 | −0.389 | −0.134 | 1.4325 | 1.2270 | 1.813 | 0.856 |
| | | 1.25 | 20 | 6.8502 | 3.2973 | 10.872 | 0.341 | 0.481 | 5.9727 | 0.2540 | 0.0645 | −0.853 | 0.327 | 6.8502 | 3.2973 | 0.399 | 0.481 |
| $c \cdot 10^{-4}$ | 15Cr2MoVa | 1.00 | 20 | 5.7665 | 3.5335 | 12.485 | 0.889 | 0.613 | 4.8351 | 0.2679 | 0.0718 | 0.010 | 0.391 | 5.7665 | 3.5335 | 1.039 | 0.613 |
| | | 1.12 | 40 | 8.4257 | 5.8851 | 34.634 | 1.081 | 0.698 | 6.5179 | 0.3335 | 0.1112 | −0.345 | 0.409 | 8.4257 | 5.8851 | 1.167 | 0.698 |
| | | 1.25 | 20 | 14.730 | 7.1011 | 50.426 | 0.344 | 0.482 | 13.010 | 0.2309 | 0.0533 | 0.343 | 0.207 | 14.730 | 7.1011 | 0.402 | 0.482 |
| $\bar{S}_T$ | 15Cr2MoVa | | 80 | 1.2500 | 0.492 | 0.0024 | −0.493 | 0.039 | 1.2491 | 0.0169 | 0.0003 | −0.369 | 0.175 | 1.2500 | 0.0492 | −0.577 | 0.039 |
| | C45 | | 80 | 1.0500 | 0.0196 | 0.0004 | 0.185 | 0.018 | 1.0578 | 0.0079 | 0.000 | 0.104 | 0.242 | 1.0500 | 0.0196 | 0.217 | 0.018 |
| | D16T1 | | 20 | 1.6540 | 0.1034 | 0.0107 | 0.534 | 0.062 | 1.6509 | 0.0267 | 0.0007 | 0.443 | 0.122 | 1.6540 | 0.1034 | 0.625 | 0.064 |
| $\alpha$ | D16T1 | 1.15 | 20 | 0.5595 | 0.0949 | 0.0090 | 0.147 | 0.169 | 0.5518 | 0.0746 | 0.0056 | −0.168 | −0.289 | 0.5595 | 0.172 | 0.172 | 0.169 |

**Table A2.** The range of the confidence intervals of cyclic properties.

| Material | $\bar{\sigma}_0$ | p | $A_1$ | | $A_{1f}$ | | $A_2$ | | $A_{2f}$ | | $\beta \cdot 10^{-3}$ | | $c \cdot 10^{-4}$ | | $\bar{S}_T$ | | $\alpha$ | |
|---|---|---|---|---|---|---|---|---|---|---|---|---|---|---|---|---|---|---|
| | | | $x_p^U$ | $x_p^L$ | $x_p^U$ | $x_p^L$ | $x_p^U$ | $x_p^L$ | $x_p^U$ | $x_p^L$ | $x_p^U$ | $x_p^L$ | $x_p^U$ | $x_p^L$ | $x_p^U$ | $x_p^L$ | $x_p^U$ | $x_p^L$ |
| 15Cr2MoVa | 1.00 | 0.01 | 0.1497 | −1.4552 | 0.6080 | 0.0889 | −0.1288 | −1.4356 | 0.6233 | 0.1060 | 0.1252 | 0.0968 | 3.6242 | 2.9893 | 1.1486 | 1.1223 | - | - |
| | | 0.50 | 2.4702 | 1.1647 | 1.6444 | 1.1254 | 2.4803 | 1.1735 | 1.6562 | 1.1389 | 0.2092 | 0.1618 | 5.3238 | 4.3912 | 1.2631 | 1.2368 | - | - |
| | | 0.99 | 5.0902 | 3.7847 | 2.6807 | 2.1617 | 5.0894 | 3.7826 | 2.6890 | 2.1717 | 0.3498 | 0.2705 | 7.8204 | 6.4503 | 1.3775 | 1.3512 | - | - |
| | 1.12 | 0.01 | 0.3589 | 0.0082 | 0.4536 | 0.1807 | 0.3505 | −0.0115 | 0.4538 | 0.1803 | 0.4806 | 0.3591 | 3.9686 | 3.2527 | 1.1486 | 1.1223 | - | - |
| | | 0.50 | 1.4091 | 1.0583 | 1.2706 | 0.9977 | 1.4342 | 1.0722 | 1.2725 | 0.9991 | 1.1503 | 0.8594 | 7.1994 | 5.9007 | 1.2631 | 1.2368 | - | - |
| | | 0.99 | 2.4598 | 2.1085 | 2.0876 | 1.8147 | 2.5179 | 2.1559 | 2.0911 | 1.8227 | 2.7523 | 2.0569 | 13.060 | 10.704 | 1.3775 | 1.3512 | - | - |
| | 1.25 | 0.01 | 0.5173 | 0.2673 | 0.1638 | −0.0919 | 0.5495 | 0.3035 | 0.1659 | −0.1022 | 4.6094 | 3.8771 | 10.509 | 9.1092 | 1.1486 | 1.1223 | - | - |
| | | 0.50 | 1.0116 | 0.7665 | 0.6744 | 0.4187 | 1.0406 | 0.7945 | 0.7015 | 0.4333 | 6.5124 | 5.4777 | 13.981 | 12.118 | 1.2631 | 1.2368 | - | - |
| | | 0.99 | 1.5156 | 1.3906 | 1.1852 | 0.9294 | 1.5317 | 1.2857 | 1.2369 | 0.9687 | 9.1981 | 7.7367 | 18.600 | 16.122 | 1.3775 | 1.3512 | - | - |
| C45 | 1.00 | 0.01 | 0.6639 | 0.4075 | 0.5365 | 0.4145 | 0.6694 | 0.4135 | 0.5353 | 0.4061 | - | - | - | - | 1.0095 | 0.9990 | - | - |
| | | 0.50 | 1.1759 | 0.9195 | 0.7801 | 0.6581 | 1.1803 | 0.9244 | 0.7933 | 0.6641 | - | - | - | - | 1.0552 | 1.0447 | - | - |
| | | 0.99 | 1.6879 | 1.4315 | 1.0236 | 0.9016 | 1.6912 | 1.4353 | 1.0513 | 0.9221 | - | - | - | - | 1.1008 | 1.0903 | - | - |
| | 1.25 | 0.01 | 0.6908 | 0.6016 | 0.5061 | 0.4317 | 0.6974 | 0.6076 | 0.5146 | 0.4404 | - | - | - | - | 1.0095 | 0.9990 | - | - |
| | | 0.50 | 0.9581 | 0.8688 | 0.7288 | 0.6544 | 0.9659 | 0.8762 | 0.7366 | 0.6624 | - | - | - | - | 1.0552 | 1.0447 | - | - |
| | | 0.99 | 1.2253 | 1.1361 | 0.9515 | 0.8771 | 1.2346 | 1.1448 | 0.9586 | 0.8844 | - | - | - | - | 1.1008 | 1.0903 | - | - |
| | 1.50 | 0.01 | 0.7166 | 0.6032 | 0.6138 | 0.4956 | 0.7287 | 0.6151 | 0.6325 | 0.5160 | - | - | - | - | 1.0095 | 0.9990 | - | - |
| | | 0.50 | 0.9432 | 0.8298 | 0.8499 | 0.7317 | 0.9555 | 0.8419 | 0.8651 | 0.7486 | - | - | - | - | 1.0552 | 1.0447 | - | - |
| | | 0.99 | 1.1698 | 1.0564 | 1.0860 | 0.9678 | 1.1822 | 1.0686 | 1.0977 | 0.9812 | - | - | - | - | 1.1008 | 1.0903 | - | - |
| D16T1 | 1.15 | 0.01 | 0.2234 | −0.0342 | 0.3211 | 0.0492 | - | - | - | - | - | - | - | - | 1.4737 | 1.3533 | 0.3941 | 0.2835 |
| | | 0.50 | 0.7380 | 0.4803 | 0.8640 | 0.5921 | - | - | - | - | - | - | - | - | 1.7142 | 1.5938 | 0.6148 | 0.5042 |
| | | 0.99 | 1.2525 | 0.9948 | 1.4068 | 1.1349 | - | - | - | - | - | - | - | - | 1.9546 | 1.8342 | 0.8355 | 0.7249 |

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
