# Peer review of "Statistical Estimation of Resistance to Cyclic Deformation of Structural Steels and Aluminum Alloy"

_metals, doi:10.3390/met12010047_

Round 1
Reviewer 1 Report
Resistance to cyclic loading is a key property of the material that determines the operational reliability of the structures. In this paper, the authors determined the dependence of the parameters of the low cycle fatigue curves on the type of load and material properties, and performed a statistical evaluation of the parameters of the low cycle loading curves for structural materials. The paper is well written and the conclusions are useful to the communications of alloy materials. The paper can be accepted after the typing errors are corrected.
(1) There is no Section 4 between Section 3 (Results and Discussion) and Section 5 (Conclusions). Please correct the numbers.
(2) There is no Table 4&5 between Table 3 (Goodness-of-Fit estimation of cyclic properties…) and Table 6 (Probabilistic values of the parameters…). Please correct the numbers.
(3) Misuses of “a” and “alpha” in Equations 4 & 5.
(4) In the deviation of equation 4, delta k should be delta 1 on the right side.
Reviewer 2 Report
The manuscript is devoted to a rather relevant topic of statistical assessment of cyclic deformation resistance. The article is quite adequately structured. The review part of the manuscript allows assessing the relevance of the work, and the review of the used references covers the scientists from completely different regions of the world.
In general, the article is quite relevant, but in my opinion a few revisions would allow slightly improving its readability for readers.
In table 3, the authors presented data on ?̅0 for three different materials under study. There are no vertical lines, separating the different values of ?̅0 in the table. To unambiguously identify what values of ?̅0 belong to what material, it would be optimal to draw vertical lines for the first two rows of the table.
The selected materials: alloy steel, carbon steel and aluminum alloy have a completely different structure and phase composition. These peculiarities will certainly influence the process of accumulation of plastic deformation.
The first paragraph of section 3.1 provides data on the cyclic properties of the materials under study. It would be worth giving the following information in this section:
- for what reasons such heterogeneous materials are compared,
- how much their structure will influence the parameters under study,
- why and how appropriate such comparison of these materials is.
Round 2
Reviewer 2 Report
The authors have finalized the article. The article can be published in a revised form.